# CREAT3R: CONFIDENCE REAGGREGATION FOR EXPLORATION-AWARE ACTIVE 3D RECONSTRUCTION

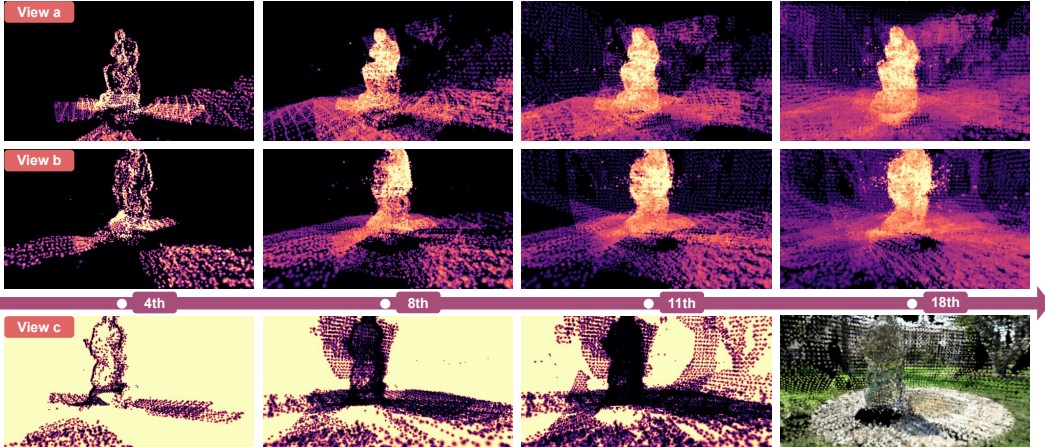

Figure 1: The proposed Creat3r progressively refines scene geometry and confidence field across successive selection rounds. The top two rows respectively illustrate the progression of the *confidence maps* of views $a$ and $b$. The bottom row shows the *exploration map* of view $c$ alongside the reconstructed geometry, which becomes increasingly detailed as more images are incorporated. At each iteration, the confidence and exploration maps jointly quantify the exploration measure of every candidate view. The exploration map highlights unobserved or weakly constrained regions (bright) to drive exploration, whereas the confidence map quantifies the reliability of reconstructed points, enabling refinement in uncertain yet already observed areas.

## ABSTRACT

We introduce *Creat3r*, an active view selection framework designed for efficient and high-quality 3D reconstruction using a limited subset of image-pose pairs. Given an initial set of selected views, our method iteratively identifies the most informative candidate views to maximize reconstruction accuracy while adhering to computational constraints. Our approach begins by generating an intermediate 3D point cloud through dense pixel correspondences and stereo triangulation, refining point estimates via the Direct Linear Transform (DLT). To assess reconstruction reliability, we introduce a *3D confidence field* that integrates camera support and view consistency, enabling a quantitative evaluation of point quality. This confidence information is then propagated to all candidate views using an efficient Gaussian projection technique, generating *2D confidence and exploration maps* for each potential viewpoint. We define an exploration measure based on these maps to evaluate and optimally select the next best view. By balancing exploration, reconstruction accuracy, and computational efficiency, Creat3r is well-suited for applications in autonomous 3D scanning, robotic vision, and multi-view scene reconstruction. To demonstrate its effectiveness, our method is evaluated against baselines using the standard 3DGS representation for 3D reconstruction from the selected views. The experimental results show that our method excels in novel view synthesis and surface reconstruction, achieving significant improvements in SSIM and F1-score.

# 1 INTRODUCTION

Recent advances in 3D Gaussian Splatting (3DGS) have enabled high-quality, real-time novel view synthesis from multi-view images. By representing a scene with a set of differentiable 3D Gaussians, 3DGS has become a leading method for immersive and realistic scene rendering. However, its state-of-the-art performance remains critically dependent on a dense set of input views, often requiring hundreds of images for a single scene. This reliance creates a significant practical bottleneck, leading to heavy computational costs, protracted optimization times, and intensive manual data acquisition, particularly for large-scale environments.

To alleviate this data acquisition burden, *active view selection* has emerged as a promising solution, aiming to intelligently choose a minimal yet informative subset of views/images. Existing methods for neural rendering typically operate in an iterative loop: they optimize the scene representation with a small set of known views, estimate uncertainty for unobserved regions, and select the "next best view" to reduce this uncertainty. For example, methods like FisherRF (Jiang et al., 2024) and Lyu et al. (2024) leverage Fisher information or variational inference to estimate model uncertainty.

While effective, these approaches suffer from two major limitations. First, they are fundamentally tied to the iterative optimization of the underlying 3D representation. Each selection step requires re-initializing and re-optimizing the Gaussian model, introducing computational redundancy and inefficiency. Second, most existing pipelines initialize using Structure-from-Motion (SfM) methods such as COLMAP, which are typically run over the entire candidate image pool. This practice leaks information from views that are supposed to be "unseen" and biases the evaluation of selection strategies. *As a result, existing approaches do not fully respect the principle of active selection, since they implicitly assume access to geometric priors derived from all candidate views.*

To address these limitations, we introduce Creat3r, a novel active view selection framework designed to be efficient, robust, and fully decoupled from costly 3DGS optimization. Instead of relying on a resource-intensive, full Gaussian model, we propose a lightweight, proxy 3D model composed of spherical Gaussians. This proxy model is not optimized iteratively but is instead built on a more fundamental and robust geometric representation of the scene.

Creat3r operates through two key mechanisms. It incrementally estimates a robust geometry of the scene by establishing pairwise correspondences between known and candidate views, then triangulating them into 3D points via Direct Linear Transformation (DLT). This procedure produces a dynamic scaffold that grows with each selection step, thereby avoiding both the leakage bias of global SfM and the instability of random initialization in sparse-view settings. Building upon this scaffold, Creat3r defines a novel exploration–exploitation criterion using two geometry-derived maps. The *confidence map* encodes the reliability of reconstructed regions, guiding refinement in uncertain but already observed areas, while the *exploration map* highlights regions that remain unobserved or poorly constrained, directing the system toward novel content. (See Figure 1.) Together, these signals balance local detail refinement with global scene expansion. The overall pipeline is illustrated in Figure 2, where geometry and confidence are re-estimated each round to guide view selection.

Through successive selections, our method produces a compact set of images sufficient for high-quality 3DGS reconstruction, while also generating a robust scene scaffold that can serve as an effective initialization for downstream tasks. We demonstrate that Creat3r consistently outperforms prior view-selection methods on both novel view synthesis and surface reconstruction tasks, achieving superior performance while significantly reducing computational and data requirements.

Our key contributions can be summarized as follows:

1. We introduce Creat3r, a novel view selection approach for 3DGS that is fully decoupled from the iterative optimization process, yielding substantial computational savings.

2. We present a robust, geometry-based initialization that incrementally builds a sparse 3D scaffold using only the selected views, thereby avoiding leakage bias caused by global SfM pipelines that are misapplied in existing approaches.

3. We propose a new exploration-exploitation selection measure that leverages a confidence map and an exploration map to intelligently guide view selection.

4. We show that Creat3r achieves state-of-the-art results for both novel view synthesis and surface reconstruction, demonstrating superior performance and data efficiency.

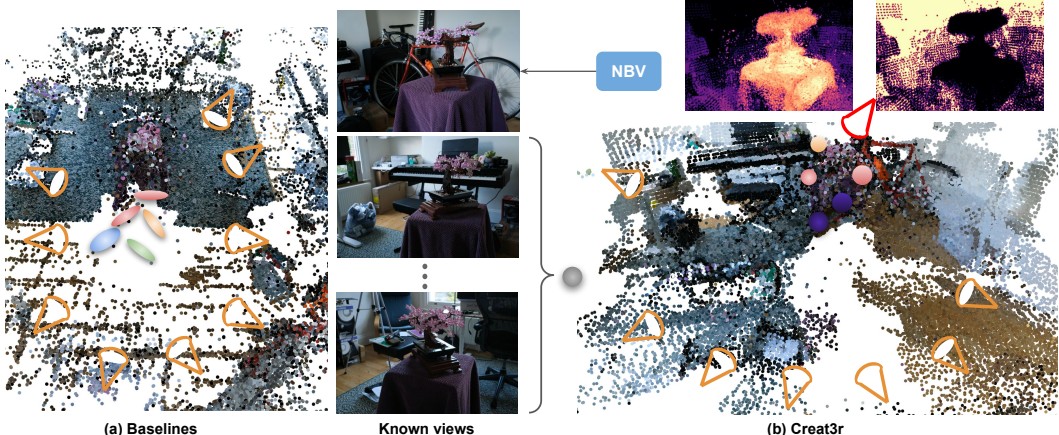

Figure 2: (a) Baseline methods re-initialize and re-optimize Gaussian ellipsoids at every iteration. Their initialization depends on the point cloud reconstructed with all the candidate views (orange cones), leading to information leakage. (b) Creat3r estimates Gaussian spheres directly from pairwise pixel correspondences. The resulting confidence field and geometry render confidence and exploration maps for each candidate view. The view with the highest exploration measure (red cone) is then selected as the *next best view* (NBV) and the newly acquired image is added to the known set.

## 2 RELATED WORK

3D reconstruction with classical structure-from-motion (SfM) (Schonberger & Frahm, 2016; Pan et al., 2024) or multi-view stereo (MVS) (Schönberger et al., 2016; Yao et al., 2018) is experiencing a renaissance with the advent of emerging radiance field (Mildenhall et al., 2021; Sun et al., 2022; Fridovich-Keil et al., 2022; Müller et al., 2022), signed distance fields (Wang et al., 2021; 2023; Li et al., 2023; Liu et al., 2023), and 3D Gaussian Splatting (3DGS) (Kerbl et al., 2023; Huang et al., 2024; Dai et al., 2024).

Active 3D reconstruction determines the next best view that will most significantly enhance the quality of the reconstruction. ActiveNeRF (Pan et al., 2022) assumes parameters to be independent and estimates the uncertainty. ActiveNeuS (Ichimaru et al., 2024) extends active selection to surface reconstruction, but only for small objects. ActiveGAMER (Chen et al., 2025) uses 3DGS and RGBD inputs for next best view selection, but only in a synthetic world. NARUTO (Feng et al., 2024) and ActiveGS (Jin et al., 2024) also take RGBD inputs and extend to the real environment.

The method of Kopanas & Drettakis (2023) samples points in space and models the point camera relationship to select new views. They have adopted InstantNGP (Müller et al., 2022) to reconstruct the scene and place cameras in the empty space. FisherRF (Jiang et al., 2024) quantifies the uncertainty of each candidate using Fisher information. It uses Laplace's approximation and computes Jacobians instead of the Hessian matrix. It modifies the rasterize function in 3DGS to speed up computation. Similar to FisherRF, the method of Goli et al. (2024) also uses Laplace's approximation for uncertainty computation in NeRF. Like FisherRF, GauSS-MI (Xie et al., 2025) also uses information gains for view selection. Their computation does not involve known views, resulting in constant search time.

The manifold sampling technique proposed by Lyu et al. (2024) takes a different approach and uses variational inference to model the parameter distribution of Gaussian primitives. They find an effective low-dimensional manifold that can speed up computation and a differentiable scheme to optimize uncertainty. Also relying on variance, the method of Ewen et al. (2025) computes pixel-wise higher moments for each candidate. For selection, they compute the variance for each candidate.

Unlike the methods mentioned above, we do not rely on optimization of 3DGS or NeRF during view selection. Instead, we use 2D correspondence predictions (Lindenberger et al., 2023; Sarlin et al., 2020; Sun et al., 2021; Leroy et al., 2024) to estimate robust geometry for active view selection.

## 3 METHOD

Consider a collection of image–pose pairs $\mathcal{S} = \{(I, W)\}$ representing a given scene or object, where each image $I$ is associated with its corresponding camera pose $W$. Selecting a view therefore entails including both its image and pose. Our goal is to develop an *active view selection* framework that enables high-quality 3D reconstruction while relying on only a limited subset of these pairs.

We initialize with a small set of *known* views, $\mathcal{S}^K = \{(I^K, W^K)\}$, which serve as the starting point for reconstruction. The remaining *candidate* views form the set $\mathcal{S}^C$, such that the entire dataset is partitioned as $\mathcal{S} = \mathcal{S}^K \dot\cup \mathcal{S}^C$. The objective is to iteratively select the most "informative" views from $\mathcal{S}^C$ to expand $\mathcal{S}^K$, while respecting constraints on the number of views and optimization steps allowed for active 3D reconstruction.

### 3.1 ROBUST POINT ESTIMATION

During active view selection, performing 3D reconstruction using the known views $\mathcal{S}^K$ yields an intermediate estimate of 3D point cloud $\tilde{\mathcal{P}}$, providing insight into the expected quality of the final reconstruction. However, reconstruction quality is not just the goal of our method—it also plays a critical role in shaping the effectiveness of the view selection criteria, as elaborated later.

Given the known set $\mathcal{S}^K = \{(I^K, W^K)\}$, we leverage correspondence networks such as Light-Glue (Lindenberger et al., 2023) to extract pixel correspondences between all pairwise views in $\mathcal{S}^K$. These correspondences are triangulated to recover 3D points, explicitly incorporating stereo geometry for accurate predictions. More formally, consider two images $I_a$ and $I_b$ in $\mathcal{S}^K$, captured by cameras $a$ and $b$, respectively. The network identifies a set of corresponding pixels, denoted as

$$\{(u_a, v_a) \leftrightarrow (u_b, v_b)\}. \tag{1}$$

For camera $a$ with projection matrix $\mathbf{P}^a$, the 3D-to-2D projection equation can be expressed as

$$\kappa_a \cdot \begin{bmatrix} u_a \\ v_a \\ 1 \end{bmatrix} = \begin{bmatrix} \mathbf{P}^a_{11} & \mathbf{P}^a_{12} & \mathbf{P}^a_{13} & \mathbf{P}^a_{14} \\ \mathbf{P}^a_{21} & \mathbf{P}^a_{22} & \mathbf{P}^a_{23} & \mathbf{P}^a_{24} \\ \mathbf{P}^a_{31} & \mathbf{P}^a_{32} & \mathbf{P}^a_{33} & \mathbf{P}^a_{34} \end{bmatrix} \begin{bmatrix} x \\ y \\ z \\ 1 \end{bmatrix}. \tag{2}$$

where $\kappa_a$ is a scalar depth factor. The same formulation applies to camera $b$ by substituting its corresponding projection matrix $\mathbf{P}^b$.

To eliminate the depth scale factor $\kappa$, we construct a linear system by multiplying the third row by $u$ and $v$, and subtracting it from the first and second rows, respectively, for both cameras. This results in the following system: $\mathbf{A}[x, y, z]^T = \mathbf{b}$, where

$$\mathbf{A} = \begin{bmatrix} u_a\mathbf{P}^a_{31} - \mathbf{P}^a_{11} & u_a\mathbf{P}^a_{32} - \mathbf{P}^a_{12} & u_a\mathbf{P}^a_{33} - \mathbf{P}^a_{13} \\ v_a\mathbf{P}^a_{31} - \mathbf{P}^a_{21} & v_a\mathbf{P}^a_{32} - \mathbf{P}^a_{22} & v_a\mathbf{P}^a_{33} - \mathbf{P}^a_{23} \\ u_b\mathbf{P}^b_{31} - \mathbf{P}^b_{11} & u_b\mathbf{P}^b_{32} - \mathbf{P}^b_{12} & u_b\mathbf{P}^b_{33} - \mathbf{P}^b_{13} \\ v_b\mathbf{P}^b_{31} - \mathbf{P}^b_{21} & v_b\mathbf{P}^b_{32} - \mathbf{P}^b_{22} & v_b\mathbf{P}^b_{33} - \mathbf{P}^b_{23} \end{bmatrix} \quad \text{and} \quad \mathbf{b} = \begin{bmatrix} \mathbf{P}^a_{14} - u_a\mathbf{P}^a_{34} \\ \mathbf{P}^a_{24} - v_a\mathbf{P}^a_{34} \\ \mathbf{P}^b_{14} - u_b\mathbf{P}^b_{34} \\ \mathbf{P}^b_{24} - v_b\mathbf{P}^b_{34} \end{bmatrix}. \tag{3}$$

We solve for $[x, y, z]^T$ using the Direct Linear Transform (DLT): $[x, y, z]^T = (\mathbf{A}^T\mathbf{A})^{-1}\mathbf{A}^T\mathbf{b}$. Applying this to all pixel correspondences across view pairs in $\mathcal{S}^K$ yields *co-visible* points whose 3D coordinates are refined under stereo geometry, ensuring robust and accurate predictions for active view selection.

### 3.2 CONFIDENCE FIELD VIA REAGGREGATION

Given the intermediate 3D reconstruction $\tilde{\mathcal{P}}$, we assign each point a confidence value to guide the selection of the next best view from $\mathcal{S}^C$. Specifically, for each predicted point $p \in \tilde{\mathcal{P}}$ with color $\mathbf{c}$ and position $\mathbf{p}$, we evaluate its visibility and consistency across the known cameras in $\mathcal{S}^K$.

For every camera $a \in \mathcal{S}^K$, point $p$ can be projected onto the image plane if it lies within the camera frustum. We define the binary *support* of camera $a$ for point $p$ as

$$g(a, p) \in \{0, 1\}, \quad g(a, p) = 1 \text{ iff } p \text{ is visible in camera } a. \tag{4}$$

Visibility alone does not guarantee reliability, as points may suffer from occlusions, clutter, or false correspondences. To capture photometric consistency, we project $p$ onto all supporting cameras and measure its color agreement with the corresponding pixels. We define the *viewing consistency* by

$$H(p) = \exp\left(-\frac{1}{\sum_{a \in \mathcal{S}^K} g(a,p)} \times \sum_{n=1}^{|\mathcal{S}^K|} g(a_n, p) \|\mathbf{c}_n - \mathbf{c}\|_2\right),\tag{5}$$

where $\mathbf{c}$ is the color of $p$, and $\mathbf{c}_n$ is the observed pixel color in camera $a_n$. If $p$ is not supported by any camera, we set $H(p) = 0$.

Finally, we define the **3D confidence field** $\mathcal{M}_{\text{Conf}}$ by weighting $H(p)$ with $\bar{g}(p)$, the fraction of known cameras in which $p$ is visible:

$$\mathcal{M}_{\text{Conf}}(p) = H(p) \times \bar{g}(p) = H(p) \times \frac{1}{|\mathcal{S}^K|} \sum_{a \in \mathcal{S}^K} g(a, p).\tag{6}$$

After each round of active view selection, we update the confidence field $\mathcal{M}_{\text{Conf}}$ for the newly constructed $\tilde{\mathcal{P}}$ by pointwise reaggregating the support and reevaluating the color consistency from all relevant cameras in $\mathcal{S}^K$.

### 3.3 View-specific confidence and exploration maps

Thus far, we have leveraged stereo geometry to improve the quality of the intermediate 3D reconstruction $\tilde{\mathcal{P}}$ and to compute its confidence field $\mathcal{M}_{\text{Conf}}$. The next step is to propagate this information to each candidate camera view in $\mathcal{S}^C$, generating corresponding 2D confidence and exploration maps. These maps provide a quantitative basis for assessing which candidate views will contribute most effectively to the next iteration of active view selection.

Inspired by 3D Gaussian Splatting (3DGS), we adopt a simplified projection scheme to transfer 3D point information to candidate views. Each point $p \in \tilde{\mathcal{P}}$ is modeled as a sphere centered at $\mathbf{p}$ with radius $r$ denoting its *influence region*. For an arbitrary position $\mathbf{x} \in \mathbb{R}^3$, the influence of $p$ is approximated using an *isotropic* Gaussian function:

$$G(\mathbf{x}) = o \cdot \exp\left(-\frac{1}{2r^2}\|\mathbf{x} - \mathbf{p}\|^2\right),\tag{7}$$

where $o$ is a constant opacity. Compared to the full 3DGS formulation, the isotropic case simplifies projection: the projected radius in the image plane is $r^{2D} = r \cdot f/\lambda$, where $f$ is the focal length and $\lambda$ denotes the depth of the Gaussian. Projected Gaussians are depth-sorted and alpha-composited to obtain the final pixel value. Leveraging this projection formulation, we determine the influence region $r$ by constraining the projected radius $r^{2D}$ in the source view to correspond to exactly one pixel.

Using $\tilde{\mathcal{P}}$ and $\mathcal{M}_{\text{Conf}}$, we generate the **2D exploration map** $\mathbf{M}_{\text{Exp}}$ and **2D confidence map** $\mathbf{M}_{\text{Conf}}$ for each candidate view. For $\mathbf{M}_{\text{Exp}}$, we assign each point a grayscale value of one, project it, and then invert the rendered image to emphasize unexplored or weakly constrained regions. For $\mathbf{M}_{\text{Conf}}$, we set the intensity of each point to its confidence value in $[0, 1]$. In both cases, we fix the opacity in Equation (7) to $o = 0.8$.

More concretely, these maps are generated for each candidate view by rendering from a simplified 3DGS model. For the exploration map, reconstructed points are modeled as small spheres, projected with maximum grayscale intensity (white), and blurred by the Gaussian kernel in Equation (7). The rendered image is then inverted so that unobserved or weakly constrained regions appear bright. The confidence map is produced analogously, except each point is colored by its confidence score rather than a uniform white value, thereby encoding the reliability of observed regions.

**Exploration measure** The exploration and confidence maps, $\mathbf{M}_{\text{Exp}}$ and $\mathbf{M}_{\text{Conf}}$, provide a quantitative way to evaluate the contribution of each candidate view in $S^C$, given the set of already selected views $S^K$. Intuitively, selecting a new view reduces the unexplored content of nearby candidates due to overlapping coverage, while views observing disjoint regions remain more valuable for selection. Building on this intuition, we define the *exploration measure* for a candidate view $(I, W) \in S^C$ as

$$\text{Exploration}(W) = \sum \mathbf{M}_{\text{Exp}}(W) - \tau \cdot \overline{\mathbf{M}}_{\text{Conf}}(W),\tag{8}$$

Table 1: Novel view synthesis evaluation on Mip-NeRF 360 dataset. We present the evaluations of 3DGS optimized with 10 and 20 selected views. (*) denotes methods initialized with COLMAP-induced subsampled points. (‡) indicates methods initialized with Creat3r-LightGlue points, and (†) indicates methods initialized with Creat3r-MASt3R. Best and second-best results are highlighted.

| Method | 10 cameras | | | 20 cameras | | |
|---|---|---|---|---|---|---|
| | PSNR↑ | SSIM↑ | LPIPS↓ | PSNR↑ | SSIM↑ | LPIPS↓ |
| FPS* | 12.529 | 0.261 | 0.613 | 14.918 | 0.389 | 0.528 |
| FPS† | 13.560 | 0.362 | 0.555 | 14.940 | 0.439 | 0.514 |
| FisherRF (Jiang et al., 2024)* | 12.625 | 0.264 | 0.608 | 15.434 | 0.390 | 0.515 |
| FisherRF (Jiang et al., 2024)† | 14.196 | 0.392 | 0.546 | 16.028 | 0.474 | 0.491 |
| Kopanas & Drettakis (2023)* | 13.022 | 0.284 | 0.596 | 15.658 | 0.407 | 0.506 |
| Kopanas & Drettakis (2023)† | 13.677 | 0.383 | 0.546 | 15.727 | 0.470 | 0.499 |
| Lyu et al. (2024)* | 12.561 | 0.264 | 0.612 | 15.446 | 0.401 | 0.518 |
| Lyu et al. (2024)† | 14.282 | 0.377 | 0.547 | 16.264 | 0.503 | 0.480 |
| Creat3r ‡ | 16.040 | 0.449 | 0.536 | 19.637 | 0.567 | 0.443 |
| Creat3r † | 17.809 | 0.511 | 0.523 | 20.678 | 0.601 | 0.397 |

where the first term quantifies the total unexplored regions visible from $W$, the second term penalizes views with high average confidence (i.e., already well-covered), and $\tau$ is a scaling factor that balances the two terms. The next best view is then selected by maximizing this exploration measure:

$$(I^*, W^*) = \arg\max_{(I,W) \in S^C} \text{Exploration}(W). \tag{9}$$

## 4 EXPERIMENT

To evaluate our method, we perform comprehensive comparisons on 3D reconstruction. The first task is novel view synthesis. The experimental setting follows previous methods, and the results are detailed in Section 4.2. The second task is surface reconstruction. This is a more severe task and has not been discussed by previous active view-selection methods. It is shown that our method is capable of reconstructing the surface under limited views. Further discussion is presented in Section 4.3.

### 4.1 IMPLEMENTATION DETAILS

Creat3r uses pixel correspondences to estimate robust geometry. Any correspondence estimation method can be used in our framework. In the experiment, we report the evaluation with two different correspondence estimation methods, LightGlue and MASt3R, for novel view synthesis. Note that we simply treat MASt3R as a correspondence network for pixel matching. We do not use their point estimate in the entire process of view selection. Our framework is 3D model agnostic, meaning it can accept any 3D reconstruction technique. For a fair comparison, we use 3DGS as our 3D representation method, following the baselines. Since Creat3r provides a reliable 3D scaffold, 3DGS converges in a very short time: We finish 3DGS optimization in 5,000 iterations. Computational efficiency is discussed in the appendix, and Table 5 shows the average selection time per iteration.

We compare Creat3r with the previous state of the arts. Specifically, we consider FisherRF (Jiang et al., 2024), Lyu et al. (2024) and Kopanas & Drettakis (2023) as competitive counterparts. Since the method of Kopanas & Drettakis (2023) is originally designed for NeRF, we adapt their method to 3DGS. We also use the 3DGS version of FisherRF in a single selection manner. All of the baseline methods have unified searching and optimization iterations. We follow the setting of Lyu et al. (2024), which takes 20,000 iterations for searching. They take another 10,000 iterations for final optimization. In addition to the uncertainty estimation approaches, we construct a simple baseline through farthest point sampling (FPS). This method only considers the position of each camera and collects the views with the largest inner distances. Despite the absence of camera orientation, this strategy can be useful, especially in an inward-captured dataset.

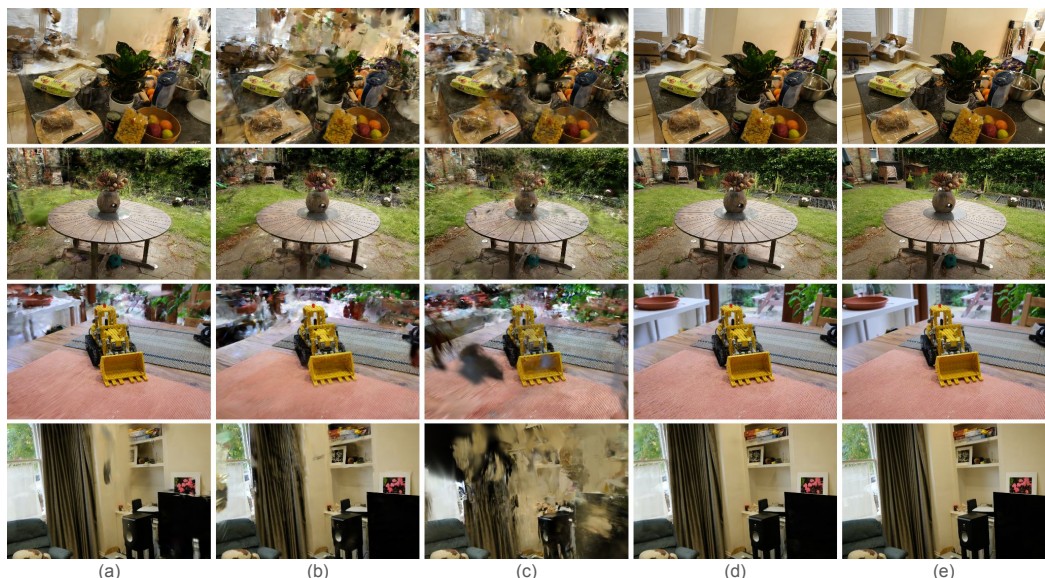

(a)  (b)  (c)  (d)  (e)

Figure 3: Qualitative comparison of active view selection on the Mip-NeRF 360 dataset for 20 selected views. The results demonstrate novel view renderings produced by competing methods: (a) FisherRF (Jiang et al., 2024), (b) Kopanas & Drettakis (2023), (c) Lyu et al. (2024), and (d) our proposed method, Creat3r. Column (e) serves as the ground truth novel view reference. The comparison highlights the superior ability of Creat3r to synthesize high-fidelity views with finer details and fewer artifacts.

## 4.2 Novel View Synthesis

The evaluation process for novel view synthesis includes several steps. First, each method selects a certain number of views for 3DGS optimization. The optimized model renders images in novel poses. The evaluation compares the image quality between rendered and ground-truth images. Standard metrics include peak signal-to-noise ratio (PSNR), structural similarity index (SSIM), and learned perceptual image patch similarity (LPIPS). The three metrics reflect different perspectives of image quality. When an image has better quality, it should have higher PSNR and SSIM, and also lower LPIPS. In this task, we use the popular MipNeRF-360 dataset for evaluation. The dataset contains nine real-world scenes, including indoor and outdoor captures. To fully expose the data efficiency of each method, the number of selections is set to 10 and 20, respectively. In the original data split, each scene has hundreds of views for optimization. Here, the training set is used as the candidate pool to find the optimal selections.

In previous literature, the evaluation process includes the sparse reconstruction from SfM. As mentioned earlier, the point cloud is reconstructed from hundreds of views that are actually treated as candidates during the selection. Using this point cloud for 3DGS initialization would reveal geometric information and lead to unfair/biased comparisons. To develop a fair comparison, we provide three different kinds of initialization. The first one is subsampling initialization. To prevent unlimited space sampling, we use the extreme values of the SfM point cloud coordinates (induced by COLMAP) as boundaries and sample within them. We are also interested in the case where other competing methods have the same initial points as ours. The second and third initializations share the same point sets estimated with Creat3r and initial views. Each of them relates to LightGlue and MASt3R-matching, respectively.

The evaluation results are listed in Table 1. A more complete comparison can be found in the Table 6. We use (*) to indicate that the methods use COLMAP-induced sampling initialization. (‡) and (†) indicate the methods start with robust points generated with Creat3r and initial views. Each baseline methods report two results with COLMAP-induced subsampling and Creat3r-MASt3R-matching initialization. Due to space limit, we report the baseline results with Creat3r-LightGlue in

Figure 4: Visualization of active view selection sequences on the 'bonsai' scene. The rows correspond to (from top to bottom): Creat3r, Lyu et al. (2024), FisherRF (Jiang et al., 2024), and Kopanas & Drettakis (2023). The first three columns display the fixed initial set, adopted from the ReconFusion benchmark (Wu et al., 2024).

the appendix. Our Creat3r-MASt3R-matching initialization has consistent improvements in all the baselines, compared to the COLMAP-induced subsampling counterpart.

For different active selection strategies, FPS shows basic performance as it only considers the positions of the camera and ignores the orientations. FisherRF and Lyu et al. (2024) have similar performance. FisherRF performs better when the view is sparse. The method finds a candidate with the most information gain and achieves a significant improvement in the initial selections. The manifold sampling method of Lyu et al. (2024) performs better when there are more views. They use posterior to the scene, which is more accurate when there are more observations. The method of Kopanas & Drettakis (2023) considers the visibility and viewing directions of sampled points. Their method has better performance in indoor or area-constrained scenes. Interestingly, the method of Kopanas & Drettakis (2023) outperforms FisherRF and Lyu et al. (2024) when the initialization is COLMAP-induced subsampling, while FisherRF and Lyu et al. (2024) surpass Kopanas & Drettakis (2023) when using Creat3r estimated points as initialization. This suggests that uncertainty estimation, either with information gain or variational inference, is more beneficial from robust geometry, while Kopanas & Drettakis (2023) is less dependent on geometry.

Creat3r outperforms all baselines in all metrics, regardless of the number of selections. Due to its robust geometry, our 3D representation requires only half of the iterations for optimization. Note that optimization is difficult due to sparse views and the absence of ground-truth points. Creat3r estimates robust geometry, projects confidence and exploration maps to each candidate, and carefully selects the next best view by exploration measure. All efforts significantly improve the novel view quality. The comparison validates that our design is effective in various real-world scenes.

The qualitative results are illustrated in Figure 3. The figure demonstrates novel view renderings of four independent scenes in Mip-NeRF 360 dataset. The comparison highlights the superior ability of Creat3r to synthesize high-fidelity views with finer details and fewer artifacts. Other competing methods render with some artifacts due to a suboptimal selection set. More qualitative comparisons are shown in Figure 5 in the appendix.

The view selection sequences generated by different approaches are visualized in Figure 4. The initial set, comprising the first three views, is adopted from the ReconFusion benchmark (Wu et al., 2024). The top row of Figure 4 illustrates the selection process of Creat3r, which exhibits a spatially diverse distribution and progressively achieves comprehensive scene exploration. In contrast, the second and third rows—representing selections driven by the uncertainty estimates of Lyu *et al.* and FisherRF—reveal that while these methods explore the scene, they suffer from intermittent redundancy. Finally, the bottom row indicates that the approach of Kopanas & Drettakis results in a highly repetitive selection pattern.

### 4.3 Surface reconstruction

The task aims to reconstruct the actual surface of the scene. While 3DGS does not produce an actual surface, we adopt the mesh extraction pipeline from 2DGS (Huang et al., 2024). After optimization,

Table 2: Surface reconstruction evaluation on Tansks&Temples dataset. **Best** results are highlighted.

|  | Precision(%) | Recall(%) | F1-score(%) |
|---|---|---|---|
| FisherRF (Jiang et al., 2024) | 7.91 | 9.81 | 8.61 |
| Kopanas & Drettakis (2023) | **17.6** | 0.58 | 0.84 |
| Lyu et al. (2024) | 5.85 | 5.86 | 5.61 |
| Creat3r | 14.09 | **25.93** | **18.05** |

we render depth maps for selected views. The depths are then fused into a voxel grid using truncated signed distance fusion Curless & Levoy (1996) and extracted via marching cubes Lorensen & Cline (1998). The evaluation process densely samples the reconstructed surface and compares it against the ground truth. The predicted point is considered valid if it is within a 5-millimeter distance from the ground-truth points. The reported metrics are precision, recall, and F1-score. In this evaluation, we use the popular Tanks&Temples dataset as the benchmark. Our setting is similar to GOF (Yu et al., 2024), which samples three scenes for evaluation, including "Caterpillar", "Ignatius", and "Truck". The scenes are more difficult than Mip-NeRF 360 scenes and exhibit a wide variety of lighting conditions, such as sunshine and reflective surfaces. Each scene provides the surface ground truth of the foreground object. Only the foreground surface is evaluated. To reconstruct the surface, all the competitors must find optimal view collections of 20 views that cover most of the appearance and regional detail.

The results are shown in Table 2. Note that the numerical values are shown in percentages. Since the three scenes are outdoors and have different exposures across views, it is challenging for 3DGS to model the scene, as the optimization solely relies on appearance differences. While the methods of FisherRF, Kopanas & Drettakis (2023), and Lyu et al. (2024) depend on optimized 3DGS for view selection, they face challenges when the optimization fails. On the other hand, Creat3r is not affected by 3DGS performance. Although our method yields lower precision compared to Kopanas & Drettakis (2023), their approach suffers from extremely limited surface coverage (low recall), resulting in a compromised F1-score. In contrast, Creat3r maintains a superior balance between precision and recall.

The robust geometry enables view selection even in challenging scenes, especially for luminance variation across views. To further validate the point, we compare Creat3r with the original 3DGS. To prevent the influence of sparse views, the 3DGS optimization uses the entire training set, which includes 200 to 400 images. The comparison is listed in Table 3. The result aligns with our point. The scene "Caterpillar" is captured in the rural field. The images have severe exposure differences. Our method is not affected by this adversary. On the other hand, "Ignatius" has a sculpture in the foreground. The material of the sculpture reflects specular light, which leads to inconsistencies across views. Creat3r has the same performance with 3DGS on the "Truck" scene, while 3DGS uses 11 times more images for training. It suggests that our selection criterion is effective and provides data efficiency for 3DGS optimization.

## 4.4 ABLATION STUDY

In the ablation study, we focus on three components of Creat3r: namely, robust point estimate, confidence reaggregation, and exploration. The study excludes one component at a time, evaluating

Table 3: F1-score comparison between Creat3r and original 3DGS. Creat3r only optimizes with 20 selected views. 3DGS uses all of the training set for optimization (more than 200.)

|  | Caterpillar | Ignatius | Truck |
|---|---|---|---|
| 3DGS (Kerbl et al., 2023) | 0.08 | 0.04 | **0.19** |
| Creat3r | **0.10** | **0.25** | **0.19** |

Table 4: Ablation study of Creat3r. Influence comparison of robust point estimate, exploration, and confidence reaggregation. We use nine different scenes in Mip-NeRF 360 for evaluation.

| Point | Exploration | Confidence | PSNR↑ | SSIM↑ | LPIPS↓ |
|-------|-------------|------------|-------|-------|--------|
|  | ✓ | ✓ | 16.479 | 0.458 | 0.574 |
| ✓ |  | ✓ | 17.022 | 0.507 | 0.525 |
| ✓ | ✓ |  | 17.457 | 0.502 | 0.543 |
| ✓ | ✓ | ✓ | **17.809** | **0.511** | **0.523** |

the performance drop for the exclusion. The evaluation uses nine scenes in the Mip-NeRF 360, and the number of selections is set to 10. The comparison is shown in Table 4. The first row excludes the point estimate. Instead, we use MASt3R predicted points as an alternative. Compared to our full model in the fourth row, the robust point estimate is the most effective technique, yielding an improvement of 1.26 in PSNR. Exploration and confidence reaggregation provide different aspects of improvement. While confidence reaggregation performs better in SSIM and LPIPS metrics, the exploration has a higher performance in PSNR. In the experiment, we find that exploration performs better in constrained scenes, such as indoor environments. On the other hand, confidence performs better on outdoor scenes. With this in mind, we achieve the best of both worlds through the exploration measure and obtain an overall better performance. The qualitative evaluation of the ablated components is presented in Figure 6 (Appendix), clearly demonstrating their respective functionalities.

## 5 LIMITATIONS AND DISCUSSION

A primary limitation of Creat3r stems from its reliance on geometric co-visibility between the candidate views and the current reconstruction scaffold. While our approach proves highly effective for inward-facing (object-centric) and forward-facing trajectories, outward-facing scenarios (e.g., the 'room' scene) present a distinct challenge. In such cases, candidate views often observe disjoint regions of the scene and may share minimal overlap with the initial estimated geometry. Consequently, these views yield null confidence maps and uniformly high-intensity exploration maps. This ambiguity can inadvertently bias the exploration measure towards redundant sampling of unconstrained regions, leading to suboptimal convergence. We emphasize that this vulnerability is inherent to the active selection paradigm; all baseline methods similarly struggle to identify informative views in the absence of initial geometric overlap. To mitigate this, we implement a regularization strategy that temporarily masks such candidates from the selection pool. These views are subsequently reintroduced as the confidence field expands and sufficient geometric connectivity is established to meaningfully constrain them.

## 6 CONCLUSION

We introduced Creat3r, a novel active view selection framework for 3D reconstruction that is computationally efficient and robust. Unlike prior methods that rely on iterative optimization of the 3D representation, Creat3r is fully decoupled from this process, leading to substantial computational savings. Our method incrementally builds a robust 3D scaffold from only the selected views, effectively avoiding the information leakage bias inherent in global SfM pipelines. Through the use of confidence map and exploration map, Creat3r balances the need for local detail refinement with global scene expansion. Our approach consistently achieves state-of-the-art results for both novel view synthesis and surface reconstruction. By providing a reliable geometry, Creat3r also significantly reduces the optimization time for downstream tasks, such as 3DGS, highlighting its superior performance and data efficiency. This work represents a significant step forward in making high-quality 3D reconstruction feasible with a minimal and informative set of images.

## ETHICS STATEMENT

Our study uses only publicly available image datasets, accessed and used under the research licenses of the datasets. We conducted no interaction or intervention with individuals and accessed no private or non-public data. We have carefully considered potential impacts and do not anticipate ethical risks beyond those commonly encountered in computer vision and machine learning research.

## REPRODUCIBILITY STATEMENT

We detail the training pipeline and evaluation protocols in both the main paper and the appendix, providing explicit hyperparameters and dataset partitions to facilitate exact replication. Additionally, we outline the implementation settings and reporting conventions to ensure our results can be accurately reproduced. To further support reproducibility, we will release the complete training and evaluation code, along with runnable scripts, in the camera-ready version.

## THE USE OF LARGE LANGUAGE MODELS (LLMS)

We used LLMs to assist with (i) improving prose (grammar, flow, and clarity), (ii) reorganizing and refining section structure, captions, and titles, and (iii) generating keywords and query strings to explore related work.

For literature discovery, every citation in the paper was located through standard search engines or digital libraries and then read and verified by the authors; we did not accept model–generated references without inspection. Numerical results, comparisons, and quotes were cross–checked against the original sources.

We reviewed all text suggested by the model to ensure that the writing aligns with our intentions, thereby confirming accuracy and originality. No confidential or sensitive information was shared with the model. The authors conducted a thorough review of the final manuscript, including all tables and figures, to verify factual accuracy and completeness.

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

# A  APPENDIX

## A.1  EXPERIMENTAL DETAIL

For novel view synthesis, we use Mip-NeRF 360 dataset as the benchmark. For every scene, $1/8$ of the views are split as novel views. The remaining views form the candidate set. Each scene has three initial views as the known set. We collect the known set as described in ReconFusion (Wu et al., 2024). Like Lyu et al. (2024), we set the active selection to gather 10 and 20 views for 3DGS optimization.

For surface reconstruction, we use the Tanks&Temples dataset as the benchmark. We follow the practice of GOF (Yu et al., 2024) and sample three scenes from the dataset's training set for evaluation. For each scene, $1/8$ of the views are split as novel views. We assign three views in each scene as the initial known set. The number of view collections is set to 20.

Creat3r is data and computational efficient. The efficiency stems from two aspects. The first one is active selection efficiency. Since the selection does not include 3DGS optimization, it takes less time to find the next best view. Average selection iteration costs of various methods are listed in Table 5. Creat3r takes less than half time needed in other counterparts. The duration is measured with a single NVIDIA V100 GPU. The second attribute is the optimization efficiency. Due to the robust geometry, Creat3r only uses half of the optimization time to converge.

Table 5: Average selection iteration duration.

| Method | Time(sec) |
|---|---|
| FisherRF (Jiang et al., 2024) | 24.124 |
| Kopanas & Drettakis (2023) | 24.965 |
| Lyu et al. (2024) | 39.470 |
| Creat3r | 10.075 |

## A.2 MORE QUANTITATIVE RESULTS

The complete evaluation of novel view synthesis is listed in Table 6. For each baseline method, we evaluated with three different initialization strategies. For all the methods, initialization with Creat3r-MASt3R-matching consistently performs best among all the initializations, and Creat3r-LightGlue initialization performs better than COLMAP-induced subsampling initialization. We observe that MASt3R produces larger amounts of 2D correspondences than LightGlue, as it estimates more 3D points and is beneficial for active view selection. Creat3r $^\ddagger$ and Creat3r $^\dagger$ progressively estimate more and more points during the selection, leading to improvement by a large margin compared to other competitors.

Table 6: Complete novel view synthesis evaluation on Mip-NeRF 360 dataset. The left and right columns show evaluations of 3DGS optimized with 10 and 20 selected views. ($^*$) denotes methods initialized with COLMAP-induced subsampling points. ($^\ddagger$) indicates methods initialized with Creat3r-LightGlue and ($^\dagger$) indicates methods initialized with Creat3r-MASt3R. **Best** and second-best results are highlighted.

| Method | 10 cameras | | | 20 cameras | | |
|---|---|---|---|---|---|---|
| | PSNR ↑ | SSIM ↑ | LPIPS ↓ | PSNR ↑ | SSIM ↑ | LPIPS ↓ |
| FPS$^*$ | 12.529 | 0.261 | 0.613 | 14.918 | 0.389 | 0.528 |
| FPS$^\ddagger$ | 12.687 | 0.318 | 0.591 | 14.409 | 0.413 | 0.539 |
| FPS$^\dagger$ | 13.560 | 0.362 | 0.555 | 14.940 | 0.439 | 0.514 |
| FisherRF (Jiang et al., 2024)$^*$ | 12.625 | 0.264 | 0.608 | 15.434 | 0.390 | 0.515 |
| FisherRF (Jiang et al., 2024)$^\ddagger$ | 13.238 | 0.342 | 0.582 | 15.254 | 0.432 | 0.523 |
| FisherRF (Jiang et al., 2024)$^\dagger$ | 14.196 | 0.392 | 0.546 | 16.028 | 0.474 | 0.491 |
| Kopanas & Drettakis (2023)$^*$ | 13.022 | 0.284 | 0.596 | 15.658 | 0.407 | 0.506 |
| Kopanas & Drettakis (2023)$^\ddagger$ | 13.039 | 0.332 | 0.585 | 15.771 | 0.458 | 0.509 |
| Kopanas & Drettakis (2023)$^\dagger$ | 13.677 | 0.383 | 0.546 | 15.727 | 0.470 | 0.499 |
| Lyu et al. (2024)$^*$ | 12.561 | 0.264 | 0.612 | 15.446 | 0.401 | 0.518 |
| Lyu et al. (2024)$^\ddagger$ | 13.308 | 0.343 | 0.578 | 15.708 | 0.454 | 0.511 |
| Lyu et al. (2024)$^\dagger$ | 14.282 | 0.377 | 0.547 | 16.264 | 0.503 | 0.480 |
| Creat3r $^\ddagger$ | **16.040** | **0.449** | **0.536** | **19.637** | **0.567** | **0.443** |
| Creat3r $^\dagger$ | **17.809** | **0.511** | **0.523** | **20.678** | **0.601** | **0.397** |

## A.3 MORE QUALITATIVE RESULTS

Figure 5 presents more instances of novel view rendering with Creat3r and competing counterparts. When the selection is suboptimal, the 3D model cannot correctly render the novel views due to less exploration or a lack of finer detail. The renderings present artifacts or holes. Creat3r demonstrates renderings closest to the ground truth. Other methods have different kinds of artifacts.

## A.4 VISUALIZATION OF ABLATION STUDY

We visualize the qualitative impact of these design choices in Figure 6. Direct reliance on raw MASt3R predictions introduces geometric scale inconsistencies, yielding noisy and blurred renderings. As observed in Figure 6(b), relying solely on the confidence map biases the selection towards local detail

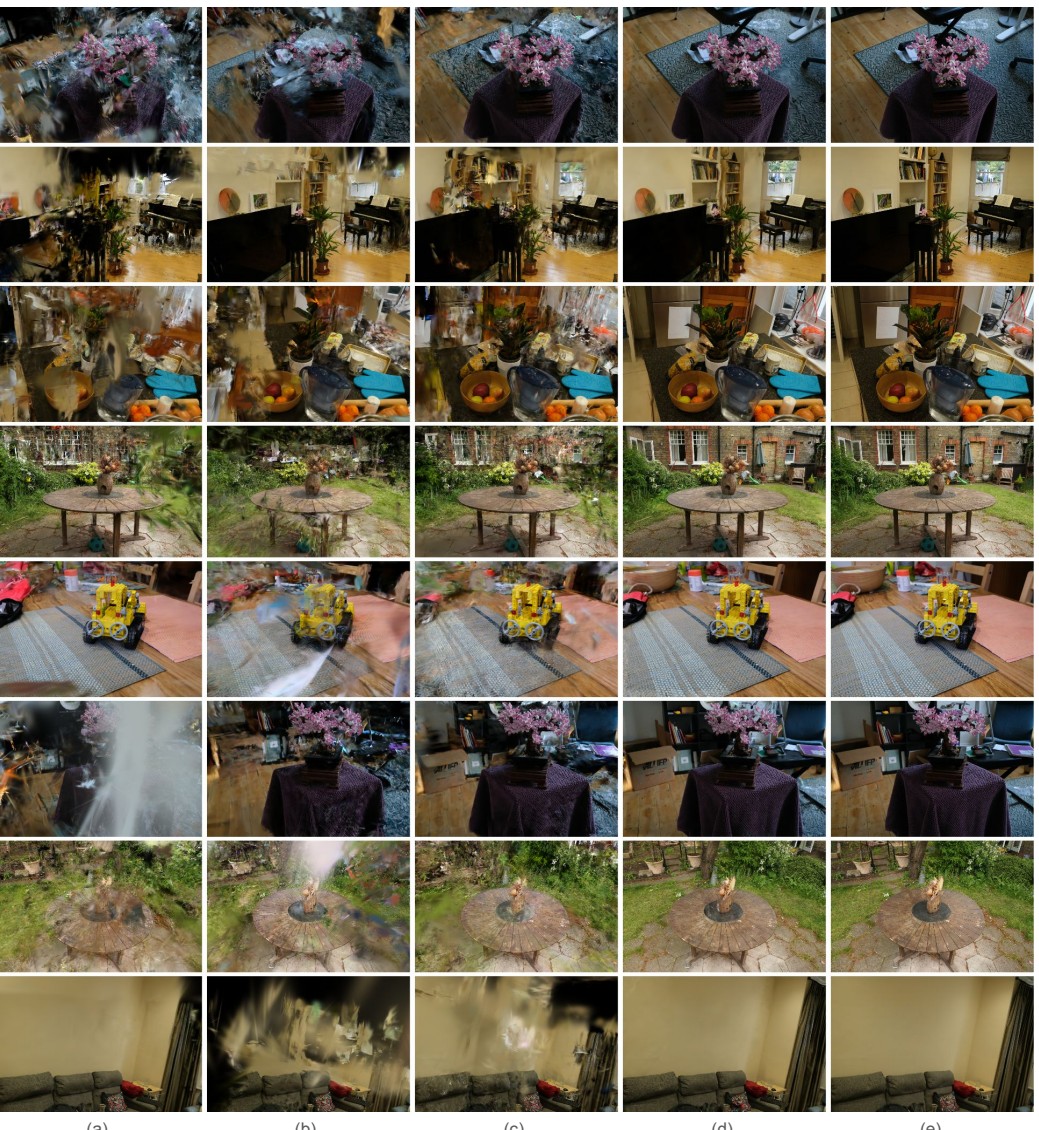

(a)    (b)    (c)    (d)    (e)

Figure 5: More qualitative comparisons of active view selection on the Mip-NeRF 360 dataset for 20 selected views. The results demonstrate novel view renderings produced by competing methods: (a) FisherRF (Jiang et al., 2024), (b) Kopanas & Drettakis (2023), (c) Lyu et al. (2024), and (d) our proposed method, Creat3r. Column (e) serves as the ground truth novel view reference. The comparison highlights the superior ability of Creat3r to synthesize high-fidelity views with finer details and fewer artifacts.

refinement; this produces high-fidelity reconstruction in observed regions (e.g., the grass) but leaves the background largely unexplored and degraded. Conversely, utilizing only the exploration map ensures broader coverage of both foreground and background but fails to resolve high-frequency details, resulting in noticeable blurring. Finally, Creat3r synergizes both exploration and confidence objectives, achieving globally consistent and highly detailed reconstructions.

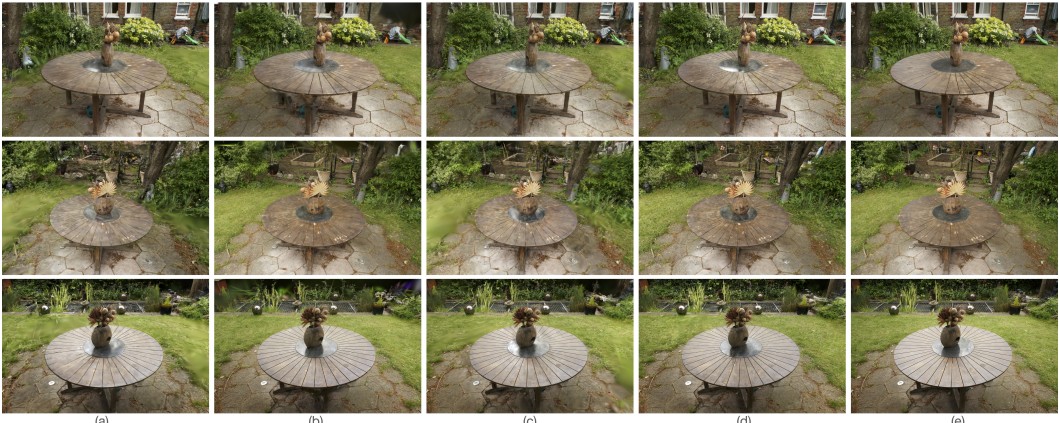

Figure 6: Qualitative evaluation of ablation components on the 'garden' scene. The rendered novel views illustrate the impact of distinct design choices: (a) substituting robust point estimation with raw MASt3R predictions, (b) relying exclusively on confidence for selection, (c) relying exclusively on exploration for selection, and (d) the full Creat3r framework. The ground truth is provided in (e). This comparison highlights the specific contribution of each component and the superior reconstruction fidelity achieved by our holistic approach.

