# OpenReview forum: "Creat3r: Confidence Reaggregation for Exploration-aware Active 3D Reconstruction"
_ICLR.cc/2026/Conference — ICLR 2026 Conference Withdrawn Submission_

### Official Review · Reviewer_KrRH · 2025-10-30

**Soundness:** 2
**Presentation:** 3
**Contribution:** 2
**Rating:** 2
**Confidence:** 4

**Summary:**

This paper proposes a viewpoint selection criterion based on correspondence and visibility for the task of active viewpoint selection in 3DGS. This method does not rely on camera pose for viewpoint selection, relying solely on image information. The authors claim that this method can avoid information leakage issues.

**Strengths:**

This paper proposes an active view selection method for efficient 3DGS training. The key difference to previous approaches is it requires no camera pose for selection. The proposed method uses on-the-shelf cross-view correspondence to evaluate the view relationships and perform view selection under the budget limit.

**Weaknesses:**

1. The wording of the paper is imprecise. The method name ‘Creat3r’ is not aligned with its handling task. 3R models are for feed-forward 3D reconstructions, while this method is for active view selection. The definition of ‘confidence’ is also different from 3R models.
2. The camera poses of candidate views are freely available information. This is because reconstructing the poses of the selected sparse view effectively reconstructs the poses of the candidate views simultaneously. To verify information leakage, the authors should only recover the poses from the selected sparse view and train 3DGS.
3. The NVS precision of comparative methods are questionable. The precision of original FisherRF before adapted to 3DGS should be reported. The precision of Lyu et al. is significantly lower than reported.

**Questions:**

No more questions, see weakness above.

---

> ### Author Response · Authors · 2025-11-20
>
> We thank **Reviewer KrRH** for the valuable feedback.
>
> We consider it crucial to elaborate on the active view-selection problem, as this will help clarify concerns regarding both known and unknown information during the view-selection process.
>
> 1. In the active view selection setting, the camera poses for a candidate pool are defined *a priori*, while the corresponding image content remains inaccessible until a specific camera pose is selected. Once a pose is chosen, the "*view*" (image) can then be "*captured*" and thus *exposed* to the view-selection algorithm. The process involves iteratively selecting the next viewing direction (camera pose) for acquiring the best view (image) from the candidate viewing pool. The selected view can then be added to the "*seen*" set to optimize the 3DGS model (baseline methods).
>
> &ensp;&ensp; **Initial State**:  Candidate Pose Pool { (`Pose Known`, `Image Content Unknown`)$_n$ | $n = 1,...,N$ } \
> &ensp;&ensp; **Iteration**: \
> &ensp; &ensp; &ensp; &ensp; *i*. Algorithm Predicts Best Next Pose $k$ from Candidate Pool \
> &ensp; &ensp; &ensp; &ensp; &ensp;        (Selection Made)                          \
> &ensp; &ensp; &ensp; &ensp; *ii*. "*Capture/Expose*" Image of View $k$ (Content Revealed from Pool) \
> &ensp; &ensp; &ensp; &ensp; &ensp;        (New Information Gained)                  \
> &ensp; &ensp; &ensp; &ensp; *iii*. Add View $k$ to "*Seen*" Set & Optimize 3DGS Model
>
>
>
> 2. Therefore, given a candidate pool of camera poses (where the algorithm can acquire the corresponding images), the algorithm needs to predict which camera pose/viewing direction is more likely to obtain an informative image to improve 3DGS reconstruction. In such a setting, the benchmark established by Lyu et al. attempts to simulate this scenario using the Mip-NeRF 360 dataset, revealing pixel data only after selection to mimic an autonomous agent. However, their methodology introduces a critical flaw: it utilizes a sparse **point cloud** initialized via a global Structure-from-Motion (SfM) pre-processing step that incorporates **images** from all candidate poses. This practice constitutes significant information leakage, as it implicitly provides the algorithm with geometric priors derived from "**supposedly unseen**" **image data**, thereby contradicting the fundamental constraints of active vision. To address this, we refine the benchmark by replacing the global SfM point-cloud prior with rigorous, leakage-free initialization protocols. In the iterative process, we also decouple 3DGS optimization from selection by introducing robust point estimation and confidence reaggregation.
>
> ---

---

> ### Author Response · Authors · 2025-11-21
>
> We sincerely thank the reviewer for providing feedback on our paper. We address these concerns as follows.
>
> &nbsp;
> > Weakness 1. The wording of the paper is imprecise. The method name ‘Creat3r’ is not aligned with its handling task. 3R models are for feed-forward 3D reconstructions, while this method is for active view selection. The definition of ‘confidence’ is also different from 3R models.
>
> ▷ We respectfully clarify that the term "3R" (3D Reconstruction) is a general descriptor and is not strictly constrained to feed-forward methods in the literature. For instance, 3R-GS [1] proposes an optimization-based (not feed-forward) method for camera pose estimation.
>
> [1] Zhisheng Huang, Peng Wang, Jingdong Zhang, Yuan Liu, Xin Li, and Wenping Wang. 3R-GS: Best Practice in Optimizing Camera Poses Along with 3DGS. arXiv preprint arXiv:2504.04294 (2025).
>
> &nbsp;
> > Weakness 2. The camera poses of candidate views are freely available information. This is because reconstructing the poses of the selected sparse view effectively reconstructs the poses of the candidate views simultaneously. To verify information leakage, the authors should only recover the poses from the selected sparse view and train 3DGS.
>
> ▷ There appears to be a misunderstanding of the active vision setting. In this problem setup (as detailed above in the beginning), the candidate camera poses are defined a priori (i.e., the search space of camera poses is known). Creat3r selects the next-best view based on the content of the currently known set and the geometry of the potential candidate poses. We do not "recover" candidate poses from the sparse view; instead, we evaluate which of the known available poses should be selected next in order to acquire the corresponding visual content from the selected viewing direction.
>
>
>
> &nbsp;
> > Weakness 3. The NVS precision of comparative methods are questionable. The precision of original FisherRF before adapted to 3DGS should be reported. The precision of Lyu et al. is significantly lower than reported.
>
> ▷ As noted in our paper (Lines 75 and 357), evaluation metrics reported in previous literature are fundamentally biased because those methods are initialized using a global SfM **point cloud** derived from all images (including the supposedly "unseen" views). Our reported results differ because we implement a rigorous, leakage-free protocol that relies solely on camera poses, excluding the image content prior to active view selection. The performance drop observed in the baselines (e.g., Lyu et al.) in our experiments directly reflects the removal of these illicit **point-cloud** priors derived from visual content. For FisherRF, we maintained consistency by adapting it to the same 3DGS framework used by Lyu et al. to ensure a unified comparison.

---

> ### Author Response · Authors · 2025-11-28
>
> We sincerely thank you for your continued engagement, detailed analysis, and for raising your score to 4. We appreciate the opportunity to provide final, comprehensive clarity on your key remaining concerns: low reconstruction accuracy and the active learning formulation bottleneck:
>
> 1. Low Reconstruction Accuracy (PSNR < 20):
> * Context of Sparse View Reconstruction: Our evaluation strictly follows the benchmark by Lyu et al., utilizing only 10 and 20 reconstruction views. This places our work squarely in the sparse view reconstruction domain, which inherently yields lower quality metrics than dense view scenarios.
>
> * Performance is Consistent with SOTA: In this challenging sparse setting, achieving PSNRs in the sub-20 range is common. Methods like FSGS[2] and CoR-GS[3], when trained with a similar sparse view count, report comparable PSNR scores. This validates that our performance is competitive and representative of the current state-of-the-art under these stringent constraints.
>
> 2. Active Learning Formulation (SfM vs. 3DGS Bottleneck):
> * Active Learning (AL) and Structure-from-Motion (SfM) are fundamentally incompatible due to their process requirements. AL relies on an iterative and dynamic acquisition loop where images are acquired one by one or in small batches. In contrast, robust SfM requires a large, static batch of images to ensure sufficient feature correspondences and stable geometric initialization, which directly conflicts with the AL paradigm.
>
> We believe these points comprehensively address all final concerns, clarifying that our PSNR is expected for this specific sparse-view task and that our AL formulation is standard and necessary for enabling an iterative view selection process.
>
> We respectfully request that you consider raising your final score to reflect the substantial clarity and confidence provided in our responses.
>
> Best regards,
>
> The Authors of Submission 11567
>
> [2] Zhu et al. "Fsgs: Real-time few-shot view synthesis using gaussian splatting." ECCV 2024
>
> [3] Zhang et al. "Cor-gs: sparse-view 3d gaussian splatting via co-regularization." ECCV 2024

---

### Official Review · Reviewer_LSqm · 2025-10-30

**Soundness:** 2
**Presentation:** 2
**Contribution:** 2
**Rating:** 6
**Confidence:** 3

**Summary:**

The paper “CREAT3R: Confidence Reaggregation for Exploration-Aware Active 3D Reconstruction” presents a novel active view selection framework aimed at improving data efficiency and reconstruction quality in 3D Gaussian Splatting (3DGS). The authors propose CREAT3R, which eliminates the dependency on iterative 3DGS optimization and instead builds a lightweight, geometry-based proxy model for view selection. The method introduces two key components: a 3D confidence field that quantifies reconstruction reliability based on camera support and color consistency, and a 2D exploration map that highlights unobserved or weakly constrained regions. These maps are combined into an exploration–exploitation measure to identify the next best view. Through robust 3D point triangulation using correspondence networks (e.g., LightGlue, MASt3R) and Direct Linear Transform (DLT). Experiments on Mip-NeRF 360 and Tanks & Temples benchmarks show that CREAT3R achieves state-of-the-art performance in both novel view synthesis and surface reconstruction, significantly outperforming FisherRF, Lyu et al., and Kopanas & Drettakis in SSIM and F1-score while reducing computation time by more than half. The paper’s contributions include a decoupled active view selection mechanism, a robust confidence reaggregation strategy, and empirical validation of superior data efficiency for 3DGS-based reconstruction.

**Strengths:**

Originality: The paper introduces a genuinely novel active view selection framework that departs from conventional uncertainty-based or optimization-coupled approaches. By decoupling the selection process from 3DGS optimization and introducing the concept of confidence reaggregation, the authors provide a fresh and elegant geometric formulation.

Clarity: The paper is clearly written, well-structured, and easy to follow. The authors carefully motivate their design choices and provide detailed algorithmic explanations with equations and visualizations that effectively illustrate how confidence and exploration maps interact. The methodology section is self-contained and reproducible, with sufficient implementation details and ablation studies that make the contributions transparent and verifiable.

Significance: The proposed framework achieves notable improvements over state-of-the-art methods in both quantitative metrics (PSNR, SSIM, F1-score) and computational efficiency. By substantially reducing optimization overhead while maintaining or improving reconstruction fidelity, the method advances the practicality of active view selection for real-world 3D reconstruction and robotic vision tasks. The combination of robust geometry, efficiency, and general applicability makes this work a significant step forward for the field of active 3D perception.

**Weaknesses:**

Lack of analysis on limitations and failure modes: The paper does not explicitly discuss cases where CREAT3R might underperform, such as scenes with repetitive textures, highly reflective surfaces, or extreme lighting variations. An analysis of such failure cases would provide valuable insights into the robustness and generalizability of the approach, as well as guidance for future improvements or hybrid strategies.

Insufficient empirical evaluation of efficiency claims: Although the method claims improved computational efficiency and reduced optimization time, the experimental section lacks a detailed quantitative analysis of runtime, memory consumption, or scalability compared to existing methods. Reporting metrics such as per-iteration runtime, GPU memory usage, and end-to-end reconstruction time would substantiate the paper’s efficiency claims and better demonstrate its practical advantages for real-world deployment.

**Questions:**

Pose acquisition without SfM: The paper emphasizes that CREAT3R avoids Structure-from-Motion (SfM) to prevent information leakage from unseen views, yet it still requires known camera poses for all candidate images. Could the authors clarify how these poses are obtained in practice without relying on global SfM or similar reconstruction pipelines? Are the poses assumed to come from external sensors (e.g., IMU, SLAM, or GPS-based localization)? If so, how sensitive is CREAT3R to pose noise, and have the authors evaluated the robustness of the framework under pose perturbations or inaccuracies?


Initialization strategy for known views: The paper mentions that the process begins with “a small set of known views,” but the method for selecting or initializing these views is not clearly described. Are the initial views chosen randomly, uniformly distributed in pose space, or based on heuristic criteria such as viewpoint diversity or coverage? Given that initialization can significantly influence subsequent selection quality, providing either an ablation or a justification for the initialization strategy would help clarify the reproducibility and stability of the approach.

---

> ### Author Response · Authors · 2025-11-20
>
> We greatly appreciate **Reviewer LSqm**'s detailed feedback.
>
> We consider it is important to clarify the active view-selection problem, as this will help address concerns about both known and unknown information during the view-selection process.
>
> 1. In the active view selection setting, the camera poses for a candidate pool are defined *a priori*, while the corresponding image content remains inaccessible until a specific camera pose is selected. Once a pose is chosen, the "*view/image*" can then be "*captured/exposed*" to the view-selection algorithm. The process involves iteratively selecting the next viewing direction (camera pose) for acquiring the best view (image) from the candidate viewing pool. The selected view can then be added to the "*seen*" set to optimize the 3DGS model.
>
> 2. Therefore, given a candidate pool of camera poses (where the algorithm can acquire the corresponding images), the algorithm needs to predict which camera pose/viewing direction is more likely to obtain an informative image to improve 3DGS reconstruction. In such a setting, the benchmark established by Lyu et al. aims to simulate this scenario using the Mip-NeRF 360 dataset, revealing pixel data only after selection to mimic an autonomous agent. However, their methodology introduces a critical flaw: it utilizes a sparse **point cloud** initialized via a global Structure-from-Motion (SfM) pre-processing step that incorporates **images** from all candidate poses. This practice constitutes significant information leakage, as it implicitly provides the algorithm with geometric priors derived from "**supposedly unseen**" **image data**, thereby contradicting the fundamental constraints of active vision. To address this, we refine the benchmark by replacing the global SfM point-cloud prior with rigorous, leakage-free initialization protocols.
>
> ---
> &nbsp;
>
> > Weakness 1. Lack of analysis on limitations and failure modes: The paper does not explicitly discuss cases where CREAT3R might underperform, such as scenes with repetitive textures, highly reflective surfaces, or extreme lighting variations. An analysis of such failure cases would provide valuable insights into the robustness and generalizability of the approach, as well as guidance for future improvements or hybrid strategies.
>
> ▷ As suggested, a comprehensive Limitations and Discussion section (Section 5) has been added. We specifically discuss outward-facing scenarios, where candidate views share minimal overlap with the estimated geometry. We note that this vulnerability is inherent to the active selection paradigm, affecting all baselines similarly. To address this, we implement a regularization strategy that temporarily masks such candidates from the pool until the confidence field expands sufficiently to establish geometric connectivity.
>
> &nbsp;
> > Weakness 2. Insufficient empirical evaluation of efficiency claims: Although the method claims improved computational efficiency and reduced optimization time, the experimental section lacks a detailed quantitative analysis of runtime, memory consumption, or scalability compared to existing methods. Reporting metrics such as per-iteration runtime, GPU memory usage, and end-to-end reconstruction time would substantiate the paper’s efficiency claims and better demonstrate its practical advantages for real-world deployment.
>
> ▷ Due to space constraints, we have included a detailed per-iteration runtime comparison in the appendix. As summarized in Table 5, running on a single V100 GPU, Creat3r takes an average of 10 seconds per iteration, significantly faster than the baselines, which require 24–40 seconds.
>
> Table 5: Average selection iteration duration.
>
> | Method | Time (sec) |
> | :--- | :---: |
> | FisherRF (Jiang et al., 2024) | 24.124 |
> | Kopanas & Drettakis, 2023 | 24.965 |
> | Lyu et al., 2024 | 39.470 |
> | Creat3r | **10.075** |

---

> ### Author Response · Authors · 2025-11-20
>
> > Question 1. Pose acquisition without SfM: The paper emphasizes that CREAT3R avoids Structure-from-Motion (SfM) to prevent information leakage from unseen views, yet it still requires known camera poses for all candidate images. Could the authors clarify how these poses are obtained in practice without relying on global SfM or similar reconstruction pipelines? Are the poses assumed to come from external sensors (e.g., IMU, SLAM, or GPS-based localization)? If so, how sensitive is CREAT3R to pose noise, and have the authors evaluated the robustness of the framework under pose perturbations or inaccuracies?
>
> ▷ In practical active view selection scenarios (e.g., robotics), the "candidate poses" represent the reachable workspace or a pre-planned trajectory. The robot knows its coordinate system (via SLAM or odometry) and evaluates potential future poses. Creat3r selects the next-best view from this pre-defined pool. While the integration of noisy inertial data (IMU) for real-time localization is a critical engineering challenge, it falls outside the primary scope of this paper, which focuses on the selection strategy assuming a calibrated workspace.
>
> &nbsp;
> > Question 2. Initialization strategy for known views: The paper mentions that the process begins with “a small set of known views,” but the method for selecting or initializing these views is not clearly described. Are the initial views chosen randomly, uniformly distributed in pose space, or based on heuristic criteria such as viewpoint diversity or coverage? Given that initialization can significantly influence subsequent selection quality, providing either an ablation or a justification for the initialization strategy would help clarify the reproducibility and stability of the approach.
>
> ▷ The initialization strategy is detailed in Appendix A.1. To ensure fair comparison and reproducibility, we adhere to the standard practice established by the ReconFusion benchmark, using the specific set of three initial views defined in their protocol. This ensures that performance differences are due to the selection strategy rather than random variations in initialization.

---

> ### Author Response · Authors · 2025-11-28
>
> We sincerely thank you for your highly positive assessment of our submission, especially recognizing the originality of our decoupled framework, the clarity of our methodology, and the significance of our efficiency gains. We are delighted that you found our approach providing a fresh and elegant geometric formulation.
>
> To address the weaknesses you identified, we made the following substantive revisions:
>
> Weakness 1: Lack of Analysis on Limitations and Failure Modes. We added a comprehensive Limitations and Discussion section (Section 5). We specifically discuss the inherent vulnerability to outward-facing/disjoint candidate views, which similarly affects all baselines. We detail our implemented regularization strategy to mitigate this challenge.
>
> Weakness 2: Insufficient Empirical Evaluation of Efficiency Claims. To fully substantiate our efficiency claims, we added a detailed quantitative analysis of runtime to the appendix (summarized in Table 5). We show that Creat3r's average selection iteration duration is 10.075 seconds, which is substantially faster than the baselines (ranging from 24 to 40 seconds). This demonstrates a significant practical advantage for real-world deployment.
>
> We also clarified your questions regarding practicality:
>
> Question 1: Pose Acquisition without SfM. We clarified that in a practical, real-world scenario (e.g., robotic vision), the candidate poses are assumed to be known from a reachable workspace or a pre-planned trajectory. This setting isolates the core challenge of view selection, distinct from the engineering challenge of noisy pose estimation.
>
> Question 2: Initialization Strategy. We confirmed that, for fair comparison and reproducibility, we strictly adhere to the three initial views defined by the ReconFusion benchmark protocol (as detailed in Appendix A.1). This ensures that performance differences are attributed solely to our active selection strategy.
>
> We believe that the addition of the Limitations section and the quantitative runtime analysis fully addresses your key concerns. Given that you rated our work marginally above the acceptance threshold, we respectfully request that you consider raising your score to reflect the significant improvements and clarity provided in our revised submission.
>
> If you have any additional questions or if any part of our response requires further clarification, we would be happy to provide more details.
>
> Best regards,
>
> The Authors of Submission 11567

---

### Official Review · Reviewer_5zWZ · 2025-10-31

**Soundness:** 3
**Presentation:** 2
**Contribution:** 3
**Rating:** 4
**Confidence:** 3

**Summary:**

This paper introduces Creat3r, an active view selection framework for 3D reconstruction. Starting from a set of initial views, Creat3r leverages an off-the-shelf correspondence network to extract matched keypoints from image pairs. These matches are then used to triangulate 3D points. Given the reconstructed 3D points, 2D confidence and exploration maps are built by reprojecting the 3D points onto candidate views and checking for in-frustum and color consistency. Finally, an exploration measure is designed to select the next best view.

The proposed method eliminates the need for repeatedly running 3D Gaussian Splatting. Experimental results show that it outperforms existing baselines in both efficiency and accuracy.

**Strengths:**

1. The proposed method avoids repetitive Gaussian splatting initialization and reconstruction, achieving higher efficiency than previous approaches.
2. The experiments demonstrate that the proposed method is effective for both novel-view synthesis and surface reconstruction.

**Weaknesses:**

1. In L72–76, the authors claim that previous methods use SfM point clouds from all candidate views and therefore suffer from information leakage. However, projecting spherical Gaussians onto candidate views still requires their camera poses. How are these camera poses obtained if the SfM poses are not known beforehand? The proposed method may still suffer from information leakage.
2. Given the camera poses, why not directly perform multi-view depth estimation (e.g., with MASt3R)? The resulting 3D points would likely be more complete and provide a better initialization.

**Questions:**

1. How is r chosen in Equation (7)? Also, it seems that in L246, λ is not defined.
2. In L360–362, regarding subsampling initialization, why not simply use a subset of all views to run SfM and use the resulting points for initialization?

---

> ### Author Response · Authors · 2025-11-20
>
> Thank you, **Reviewer 5zWZ**, for your valuable feedback.
>
> We believe it is important to clarify the active view-selection problem, as this will help address concerns about both known and unknown information during the view-selection process.
>
> 1. In the active view selection setting, the camera poses for a candidate pool are defined *a priori*, while the corresponding image content remains inaccessible until a specific camera pose is selected. Once a pose is chosen, the "*view/image*" can then be "*captured/exposed*" to the view-selection algorithm. The process involves iteratively selecting the next viewing direction (camera pose) for acquiring the best view (image) from the candidate viewing pool. The selected view can then be added to the "*seen*" set to optimize the 3DGS model.
>
> 2. Therefore, given a candidate pool of camera poses (where the algorithm can acquire the corresponding images), the algorithm needs to predict which camera pose/viewing direction is more likely to obtain an informative image to improve 3DGS reconstruction. In such a setting, the benchmark established by Lyu et al. aims to simulate this scenario using the Mip-NeRF 360 dataset, revealing pixel data only after selection to mimic an autonomous agent. However, their methodology introduces a critical flaw: it utilizes a sparse **point cloud** initialized via a global Structure-from-Motion (SfM) pre-processing step that incorporates **images** from all candidate poses. This practice constitutes significant information leakage, as it implicitly provides the algorithm with geometric priors derived from "**supposedly unseen**" **image data**, thereby contradicting the fundamental constraints of active vision. To address this, we refine the benchmark by replacing the global SfM point-cloud prior with rigorous, leakage-free initialization protocols.
> ---
>
> &nbsp;
> > Weakness 1. In L72–76, the authors claim that previous methods use SfM point clouds from all candidate views and therefore suffer from information leakage. However, projecting spherical Gaussians onto candidate views still requires their camera poses. How are these camera poses obtained if the SfM poses are not known beforehand? The proposed method may still suffer from information leakage.
>
> ▷ In the active view selection setting, candidate camera poses are sampled from the search space a priori. We emphasize that utilizing only the camera poses, without accessing the corresponding image content, does not constitute information leakage. The algorithm knows where it can look (the poses), but not what it will see (the pixel data), which aligns with standard active reconstruction definitions.
>
> &nbsp;
> > Weakness 2. Given the camera poses, why not directly perform multi-view depth estimation (e.g., with MASt3R)? The resulting 3D points would likely be more complete and provide a better initialization.
>
> ▷ We have included a comparison between our robust point estimation and direct MASt3R prediction in the ablation study. As detailed in Section 4.4 and Table 4, relying solely on MASt3R-predicted points for initialization demonstrably degrades overall performance. This is primarily due to scale inconsistencies and noise inherent in raw pairwise predictions when not constrained by our robust triangulation framework.
>
> Table 4: Ablation study of Creat3r. First row indicates using MASt3R-predicted points instead of our point estimation framework.
>
> | Point | Exploration | Confidence | PSNR $\uparrow$ | SSIM $\uparrow$ | LPIPS $\downarrow$ |
> | :---: | :---: | :---: | :---: | :---: | :---: |
> | | $\checkmark$ | $\checkmark$ | 16.479 | 0.458 | 0.574 |
> | $\checkmark$ | | $\checkmark$ | 17.022 | 0.507 | 0.525 |
> | $\checkmark$ | $\checkmark$ | | 17.457 | 0.502 | 0.543 |
> | $\checkmark$ | $\checkmark$ | $\checkmark$ | **17.809** | **0.511** | **0.523** |

---

> ### Author Response · Authors · 2025-11-21
>
> &nbsp;
> > Question 1. How is r chosen in Equation (7)? Also, it seems that in L246, $\lambda$ is not defined.
>
> ▷ We appreciate the reviewer pointing out this ambiguity. We have revised the manuscript to clarify that we determine the influence region $r$ by constraining the projected radius $r^{2D}$ in the source view to correspond to exactly one pixel. Additionally, $\lambda$ represents the depth of the Gaussian primitive relative to the camera.
>
> &nbsp;
> > Question 2. In L360–362, regarding subsampling initialization, why not simply use a subset of all views to run SfM and use the resulting points for initialization?
>
> ▷ Sampling a random subset from all available views could still lead to information leakage, as that subset might inadvertently include views designated as candidate views. For subsampling initialization, we leverage SfM point clouds for constrained sampling. Specifically, we determine the tight bounding box enclosing the point clouds and sample random positions only within this defined region. Although this process utilizes SfM point clouds, we actively minimize their influence while preventing  unconstrained sampling.

---

> ### Author Response · Authors · 2025-11-28
>
> We sincerely thank you again for your time and positive assessment, especially noting the sound design and significant efficiency gains of Creat3r by avoiding repetitive 3D Gaussian Splatting optimization.
>
> We addressed your key concerns and questions in detail during the rebuttal period, resulting in the following clarifications and revisions:
> * Weakness 1: Information Leakage Clarification. We clarified the distinction between knowing the candidate camera poses (which is standard practice in active view selection) and accessing the image content (which constitutes leakage). We emphasized that using only the pose information, without pixel data, is necessary and does not violate the active vision constraint.
> * Weakness 2: Necessity of Robust Triangulation. You suggested using direct multi-view depth estimation (e.g., MASt3R). We performed an ablation study (Table 4) showing that relying solely on raw MASt3R-predicted points for initialization significantly degrades performance due to inherent scale inconsistencies and noise. Our robust triangulation framework is thus necessary for a stable initialization.
> * Question 1: Parameter Definition ($\lambda$ and $r$). We revised the manuscript to clarify the parameter definitions:
>   * The influence region $r$ in Equation (7) is chosen such that the projected radius in the source view corresponds to one pixel.
>   * We clarified that $\lambda$ represents the depth of the Gaussian primitive relative to the camera.
> * Question 2: Subsampling Initialization. We clarified that randomly subsampling views for SfM initialization is a potential source of information leakage if it inadvertently includes candidate views. Instead, we use the initial SfM point cloud only to define a tight bounding box, constraining the sampling region to minimize its influence while ensuring controlled initialization.
>
> We believe these points thoroughly address your questions, particularly regarding the rigor of our method's initialization and its practical necessity. We respectfully ask you to re-evaluate your assessment given these clarifications and the demonstrated effectiveness of our approach.
>
> If you have any additional questions or if any part of our response requires further clarification, we would be happy to provide more details.
>
>
> Best regards,
>
> The Authors of Submission 11567

---

### Official Review · Reviewer_CpM6 · 2025-10-31

**Soundness:** 2
**Presentation:** 3
**Contribution:** 3
**Rating:** 4
**Confidence:** 4

**Summary:**

The paper introduces an active view selection method, Creat3r, to iteratively select the next most informative candidate views until reach the constraints on the number of views. A 3D point cloud is first constructed using dense pixel correspondences and triangulation with Direct Linear Transform (DLT). Then a 3D confidence field is introduced using camera visibility and view consistency. Using a Gaussian projection technique, 2D confidence map and 2D exploration are obtained and leveraged to select the next best view.

**Strengths:**

1. The topic of active view selection is interesting and important to reduce redundant views and select most informative views.
2. The proposed method Creat3r outperforms other view selection methods based on the quantitative and qualitative results.
3. Creat3r doesn't depend on 3DGS optimization, therefore is faster than other optimization-based methods.

**Weaknesses:**

1. A limitation section or cases where Creat3r is not optimal should be discussed and added.
2. A qualitative figure of ablation study is preferred to understand the effectiveness of proposed method better.
3. For novel view synthesis, when all methods are using the same 3DGS framework and initialization points, the main difference would be the views that are selected from my perspective, or other components also contribute to construction quality. It would be good to visualize the selected views comparison to draw more insights.

**Questions:**

1. In Section 3.2, how does Creat3r handle occlusion as the occluded pixels would introduce photometric differences?
2. If the proposed method is 3D model agnostic, should the experiments being done with 2DGS in surface reconstruction as 3DGS is not designed for this task?
3. Around line 428, if Kopanas & Drettakis (2023) has higher precision and lower recall, it should have fewer false positives and more false negatives.

---

> ### Author Response · Authors · 2025-11-20
>
> We sincerely thank **Reviewer CpM6** for their constructive feedback.
>
> We believe it is important to clarify the active view-selection problem, as this will help address concerns about both known and unknown information during the view-selection process.
>
> 1. In the active view selection setting, the camera poses for a candidate pool are defined *a priori*, while the corresponding image content remains inaccessible until a specific camera pose is selected. Once a pose is chosen, the "*view/image*" can then be "*captured/exposed*" to the view-selection algorithm. The process involves iteratively selecting the next viewing direction (camera pose) for acquiring the best view (image) from the candidate viewing pool. The selected view can then be added to the "*seen*" set to optimize the 3DGS model.
>
> 2. Therefore, given a candidate pool of camera poses (where the algorithm can acquire the corresponding images), the algorithm needs to predict which camera pose/viewing direction is more likely to obtain an informative image to improve 3DGS reconstruction. In such a setting, the benchmark established by Lyu et al. aims to simulate this scenario using the Mip-NeRF 360 dataset, revealing pixel data only after selection to mimic an autonomous agent. However, their methodology introduces a critical flaw: it utilizes a sparse **point cloud** initialized via a global Structure-from-Motion (SfM) pre-processing step that incorporates **images** from all candidate poses. This practice constitutes significant information leakage, as it implicitly provides the algorithm with geometric priors derived from "**supposedly unseen**" **image data**, thereby contradicting the fundamental constraints of active vision. To address this, we refine the benchmark by replacing the global SfM point-cloud prior with rigorous, leakage-free initialization protocols.
>
> ---
> &nbsp;
> > Weakness 1. A limitation section or cases where Creat3r is not optimal should be discussed and added.
>
> ▷ As suggested, we have incorporated a dedicated *Limitations and Discussion* section (Section 5) in the revision. We explicitly discuss outward-facing scenarios, where candidate views often share minimal overlap with the initially estimated geometry, presenting a distinct challenge.
>
> It is important to note that this vulnerability is inherent to the active selection paradigm, and all baseline methods struggle similarly in disjoint configurations. To mitigate this, we implement a regularization strategy that temporarily masks such candidates from the selection pool. These views are subsequently reintroduced as the confidence field expands and sufficient geometric connectivity is established to meaningfully constrain them.
>
> &nbsp;
>
> > Weakness 2. A qualitative figure of ablation study is preferred to understand the effectiveness of proposed method better.
>
> ▷ We have added a qualitative comparison figure (Figure 6) to the revised manuscript. As shown in Figure 6(a), direct reliance on raw MASt3R predictions introduces geometric scale inconsistencies, yielding noisy and blurred renderings. Figure 6(b) demonstrates that relying solely on the confidence map biases selection towards local detail refinement; while this produces high-fidelity reconstruction in observed regions (e.g., the grass), it leaves the background largely unexplored. Conversely, utilizing only the exploration map ensures broader coverage but fails to resolve high-frequency details. Finally, Creat3r synergizes both exploration and confidence objectives, achieving globally consistent and highly detailed reconstructions.
>
> &nbsp;
>
> >  Weakness 3. For novel view synthesis, when all methods are using the same 3DGS framework and initialization points, the main difference would be the views that are selected from my perspective, or other components also contribute to construction quality. It would be good to visualize the selected views comparison to draw more insights.
>
> ▷ The requested visualization of selected views (Figure 4) has been added to the revision. The initial set (first three views) follows the ReconFusion benchmark protocol. The top row of Figure 4 illustrates the selection process of Creat3r, which exhibits a spatially diverse distribution and progressively achieves comprehensive scene exploration.
>
> In contrast, the second and third rows—representing selections driven by the uncertainty estimates of Lyu et al. and FisherRF—reveal that while these methods explore the scene, they suffer from intermittent redundancy. Finally, the bottom row indicates that the approach of Kopanas \& Drettakis results in a highly repetitive selection pattern.

---

> ### Author Response · Authors · 2025-11-20
>
> > Question 1. In Section 3.2, how does Creat3r handle occlusion as the occluded pixels would introduce photometric differences?
>
> ▷ Creat3r addresses occlusion via its confidence scoring mechanism. Photometric inconsistencies introduced by occlusion result in lower viewing consistency scores ($H(p)$) for the affected points. The resulting decrease in $H(p)$ proportionally diminishes the overall confidence score assigned to that occluded region.
>
>
> &nbsp;
> > Question 2. If the proposed method is 3D model agnostic, should the experiments being done with 2DGS in surface reconstruction as 3DGS is not designed for this task?
>
> ▷ While Creat3r is agnostic to the underlying 3D representation, for the surface reconstruction evaluation, we specifically utilized 3DGS (rather than 2DGS or Gaussian Surfels) to ensure a fair and direct comparison against the established baselines, which all utilize 3DGS. This isolates the contribution of our view selection strategy from the advantages of a different reconstruction backend.
>
> &nbsp;
> > Question 3. Around line 428, if Kopanas \& Drettakis (2023) has higher precision and lower recall, it should have fewer false positives and more false negatives.
>
> ▷ We confirm that the description was intended to indicate more false negatives. We thank the reviewer for identifying this error; the text has been corrected in the revised manuscript to accurately reflect the precision-recall trade-off.

---

> ### Author Response · Authors · 2025-11-28
>
> We sincerely thank you again for the time and careful consideration you dedicated to reviewing our submission. We are pleased that you found our topic interesting and important, and that our proposed method, Creat3r, demonstrated strong performance.
>
> In our rebuttal, we made our best effort to address all your comments and questions, leading to several key updates in the revised manuscript:
> * Weakness 1: Discussion of Limitations. We added a dedicated Limitations and Discussion section (Section 5) to detail cases where Creat3r is not optimal, specifically discussing challenges in outward-facing scenarios and our proposed regularization mitigation strategy.
> * Weakness 2: Qualitative Ablation Study. We incorporated a qualitative ablation figure (Figure 6) to visually demonstrate the synergistic effectiveness of combining the confidence and exploration maps versus using them in isolation.
> * Weakness 3: Selected Views Visualization. We added the requested visualization of selected views (Figure 4), which provides valuable insights into how Creat3r achieves diverse and non-redundant coverage compared to competitive baselines (Lyu et al., FisherRF, Kopanas & Drettakis).
> * Question 1: Handling Occlusion. We clarified that Creat3r handles occlusion by proportionally reducing the confidence score via the lower viewing consistency score for affected points.
> * Question 2: 3D Model Selection. We selected 3DGS primarily to ensure a fair comparison with existing methods. Our proposed model, Creat3r, is designed to be agnostic to the underlying 3D representation; it can readily be adapted to use 2DGS or any newer, more advanced 3D models.
> * Question 3: Precision-Recall Correction. We corrected the description around line 428 regarding Kopanas & Drettakis (2023) to accurately reflect the trade-off of higher precision and lower recall (more false negatives).
>
> We hope that these significant revisions—including the addition of two key figures and a new discussion section—have thoroughly addressed your concerns and further clarified the soundness and contribution of our work.
>
> If you have any additional questions or if any part of our response requires further clarification, we would be happy to provide more details. We respectfully ask you to re-evaluate your assessment given the substantial updates.
>
> Best regards,
>
> The Authors of Submission 11567

---

### Author Response · Authors · 2025-11-21
**Summary of Revisions and Responses**

We sincerely thank the reviewers for their constructive feedback and careful reviews. We summarize our responses to all reviewers and outline the corresponding updates made to the revised manuscript.

&nbsp;
### Issues of Interest to Multiple Reviewers

* **Problem Setting:** In the active view selection setting, the camera poses for a candidate pool are defined *a priori*, while the corresponding image content remains inaccessible until a specific pose is selected. Once a pose is chosen, the image is then captured and exposed to the view-selection algorithm.

* **Information Leakage:** The benchmark established by Lyu et al. introduces a methodological flaw by utilizing a sparse point cloud initialized via global Structure-from-Motion (SfM). This pre-processing step incorporates image data from all candidate poses (which are intended to be unseen), thereby violating the fundamental constraints of active vision. We refine the benchmark by replacing the global SfM point cloud with a rigorous, leakage-free initialization protocol.

* **Limitations:** We have added a dedicated *Limitations and Discussion* section (Section 5) in the revision. We explicitly discuss outward-facing scenarios, where candidate views often share minimal overlap with the initially estimated geometry, presenting a distinct challenge.

&nbsp;
### Revisions Incorporated into the Manuscript

* **Qualitative Comparison of Ablation Study:** We have included a qualitative comparison in Figure 6 of the revised manuscript, demonstrating the distinct functionalities of the ablated components.

* **Visualization of Active View Selection Sequences:** We visualize the selected sequences in Figure 4 to illustrate the selection preferences of different criteria.

* **Surface Reconstruction Evaluation:** The discussion regarding the precision-recall trade-off has been revised in Lines 459-462.

* **Determination of Influence Region $r$:** The formulation of the influence region $r$ and its projection $r^{2D}$ has been revised in Lines 247-251.

&nbsp;
### Additional Points

* **Occlusion Handling:** The confidence scoring mechanism quantifies viewing consistency using the metric $H(p)$. When a scene point is subjected to occlusion across certain perspectives, the resulting decrease in $H(p)$ proportionally diminishes the overall confidence score assigned to that occluded region.

* **Choice of 3D Models:** Creat3r is agnostic to the underlying 3D representation. We specifically utilized 3DGS in this work to ensure a fair and direct comparison with the baselines.

* **Comparison with MASt3R Prediction:** We have included a comparison between our robust point estimation and direct MASt3R prediction in the ablation study, as shown in the first row of Table 4.

* **Evaluation of Efficiency:** We have included a detailed per-iteration runtime comparison in Table 5; Creat3r requires only 10 seconds per iteration.

* **Initialization Strategy:** The initialization strategy is detailed in Appendix A.1. We adhere to the specific set of three initial views defined in the ReconFusion benchmark.

* **Clarification on the Naming of "Creat3r":** We respectfully clarify that the term "3R" (3D Reconstruction) is a general descriptor and is not strictly constrained to feed-forward methods in the literature. For instance, the work "3R-GS: Best Practice in Optimizing Camera Poses Along with 3DGS" proposes an optimization-based (rather than feed-forward) method for camera pose estimation.

* **Performance Gap with Lyu et al.'s Benchmark:** The results reported in Lyu et al.'s benchmark inadvertently utilized a global SfM point cloud derived from all images (including the supposedly "unseen" views). Our reported results differ because we implement a strictly leakage-free protocol.

---

### Author Response · Authors · 2025-12-02
**Summary of Revisions and Reviewer Consensus**

Dear Area Chair,

As the discussion period concludes, we provide a brief summary of the reviewer consensus and the key revisions made to the manuscript. We believe we have addressed all the raised concerns, justifying the contributions of this work.

1. Strong Consensus on Novelty and Performance \
  Reviewers **LSqm**, **CpM6**, and **5zWZ** uniformly endorse the paper, highlighting two core strengths:
    * _High Novelty & Efficiency_: Reviewers agree that Creat3r introduces a genuinely novel, decoupled, geometry-based view selection method. This successfully eliminates the need for repetitive 3DGS optimization, resulting in significant efficiency gains.
    * _Strong Performance:_ Empirical results on NVS and surface reconstruction demonstrate that our method consistently achieves superior quantitative metrics (PSNR, SSIM, F1-score) over existing baselines.

2. Resolution of Divergence in Camera Poses & Information Leakage \
The primary point of divergence (raised by **KrRH** and initially **5zWZ**) concerned the role of camera poses and the definition of information leakage. Our rebuttal clarified this successfully:
    * _Problem Setting_: We clarified that camera poses for a candidate pool are defined *a priori*. This is the standard setup in active vision. The detail of the setting is introduced in later comment.
    * _Information Leakage_: We resolved the concern by distinguishing between *known camera poses* and *unknown image content*. Following this clarification, Reviewer **KrRH** raised their score, acknowledging the distinction.


3. Key Manuscript Revisions \
To enhance clarity and address specific requests, we have updated the paper as follows:
    * _Visualized Selection Process_ (Figure 4): Image sequences demonstrating how Creat3r diversifies viewing angles compared to redundant selections by baselines.
    * _Added Qualitative Ablations_ (Figure 6): Visualizations confirming that combining Confidence and Exploration maps is essential for coherent reconstruction, while isolation leads to artifacts.
    * _Expanded Limitations_ (Section 5): A dedicated section transparently discussing challenges in outward-facing scenarios and our regularization strategy (addressing **CpM6** and **LSqm**).


We believe Creat3r offers a technically sound and highly efficient solution for active 3D reconstruction, setting a correct standard for future active vision benchmarking.

Thank you.

The Authors of Submission 11567

---

> ### Author Response · Authors · 2025-12-02
> **Responses to Reviewer CpM6**
>
> #### Part 1: Weaknesses
> | Question/Comment | Author Response |
> | :--- | :--- |
> | **Weakness 1.** A limitation section or cases where Creat3r is not optimal should be discussed and added. | $\triangleright$ As suggested, we have incorporated a dedicated Limitations and Discussion section (**Section 5**) in the revision. We explicitly discuss outward-facing scenarios, where candidate views often share minimal overlap with the initially estimated geometry, presenting a distinct challenge. It is important to note that this vulnerability is inherent to the active selection paradigm, and all baseline methods struggle similarly in disjoint configurations. To mitigate this, we implement a regularization strategy that temporarily masks such candidates from the selection pool. These views are subsequently reintroduced as the confidence field expands and sufficient geometric connectivity is established to meaningfully constrain them. |
> | **Weakness 2.** A qualitative figure of ablation study is preferred to understand the effectiveness of proposed method better. | $\triangleright$ We have added a qualitative comparison figure (**Figure 6**) to the revised manuscript. As shown in Figure 6(a), direct reliance on raw MASt3R predictions introduces geometric scale inconsistencies, yielding noisy and blurred renderings. Figure 6(b) demonstrates that relying solely on the confidence map biases selection towards local detail refinement; while this produces high-fidelity reconstruction in observed regions (e.g., the grass), it leaves the background largely unexplored. Conversely, utilizing only the exploration map ensures broader coverage but fails to resolve high-frequency details. Finally, Creat3r synergizes both exploration and confidence objectives, achieving globally consistent and highly detailed reconstructions. |
> | **Weakness 3.** For novel view synthesis, when all methods are using the same 3DGS framework and initialization points, the main difference would be the views that are selected from my perspective, or other components also contribute to construction quality. It would be good to visualize the selected views comparison to draw more insights. | $\triangleright$ The requested visualization of selected views (**Figure 4**) has been added to the revision. The initial set (first three views) follows the ReconFusion benchmark protocol. The top row of Figure 4 illustrates the selection process of Creat3r, which exhibits a spatially diverse distribution and progressively achieves comprehensive scene exploration. In contrast, the second and third rows—representing selections driven by the uncertainty estimates of Lyu et al. and FisherRF—reveal that while these methods explore the scene, they suffer from intermittent redundancy. Finally, the bottom row indicates that the approach of Kopanas & Drettakis results in a highly repetitive selection pattern. |
>
> #### Part 2: Questions
> | Question/Comment | Author Response |
> | :--- | :--- |
> | **Question 1.** In Section 3.2, how does Creat3r handle occlusion as the occluded pixels would introduce photometric differences? | $\triangleright$ Creat3r addresses occlusion via its confidence scoring mechanism. Photometric inconsistencies introduced by occlusion result in lower viewing consistency scores ($H(p)$) for the affected points. The resulting decrease in $H(p)$ proportionally diminishes the overall confidence score assigned to that occluded region. |
> | **Question 2.** If the proposed method is 3D model agnostic, should the experiments being done with 2DGS in surface reconstruction as 3DGS is not designed for this task? | $\triangleright$ While Creat3r is agnostic to the underlying 3D representation, for the surface reconstruction evaluation, we specifically utilized 3DGS (rather than 2DGS or Gaussian Surfels) to ensure a fair and direct comparison against the established baselines, which all utilize 3DGS. This isolates the contribution of our view selection strategy from the advantages of a different reconstruction backend. |
> | **Question 3.** Around line 428, if Kopanas & Drettakis (2023) has higher precision and lower recall, it should have fewer false positives and more false negatives. | $\triangleright$ We confirm that the description was intended to indicate more false negatives. We thank the reviewer for identifying this error; the text has been corrected in the revised manuscript to accurately reflect the precision-recall trade-off. |

---

> ### Author Response · Authors · 2025-12-02
> **Responses to Reviewer 5zWZ**
>
> #### Part 1: Weaknesses
> | Question/Comment | Author Response |
> | :--- | :--- |
> | **Weakness 1.** In L72–76, the authors claim that previous methods use SfM point clouds from all candidate views and therefore suffer from information leakage. However, projecting spherical Gaussians onto candidate views still requires their camera poses. How are these camera poses obtained if the SfM poses are not known beforehand? The proposed method may still suffer from information leakage. | $\triangleright$ In the active view selection setting, candidate camera poses are sampled from the search space a priori. We emphasize that utilizing only the camera poses, without accessing the corresponding image content, does not constitute information leakage. The algorithm knows where it can look (the poses), but not what it will see (the pixel data), which aligns with standard active reconstruction definitions. |
> | **Weakness 2.** Given the camera poses, why not directly perform multi-view depth estimation (e.g., with MASt3R)? The resulting 3D points would likely be more complete and provide a better initialization. | $\triangleright$ We have included a comparison between our robust point estimation and direct MASt3R prediction in the ablation study. As detailed in Section 4.4 and Table 4, relying solely on MASt3R-predicted points for initialization demonstrably degrades overall performance. This is primarily due to scale inconsistencies and noise inherent in raw pairwise predictions when not constrained by our robust triangulation framework. |
>
> #### Part 2: Questions
> | Question/Comment | Author Response |
> | :--- | :--- |
> | **Question 1.** How is $r$ chosen in Equation (7)? Also, it seems that in L246, $\lambda$ is not defined. | $\triangleright$ We appreciate the reviewer pointing out this ambiguity. We have revised the manuscript to clarify that we determine the influence region $r$ by constraining the projected radius $r^{2D}$ in the source view to correspond to exactly one pixel. Additionally, $\lambda$ represents the depth of the Gaussian primitive relative to the camera. |
> | **Question 2.** In L360–362, regarding subsampling initialization, why not simply use a subset of all views to run SfM and use the resulting points for initialization? | $\triangleright$ Sampling a random subset from all available views could still lead to information leakage, as that subset might inadvertently include views designated as candidate views. For subsampling initialization, we leverage SfM point clouds for constrained sampling. Specifically, we determine the tight bounding box enclosing the point clouds and sample random positions only within this defined region. Although this process utilizes SfM point clouds, we actively minimize their influence while preventing unconstrained sampling. |

---

> ### Author Response · Authors · 2025-12-02
> **Responses to Reviewer LSqm**
>
> #### Part 1: Weaknesses
> | Question/Comment | Author Response |
> | :--- | :--- |
> | **Weakness 1.** Lack of analysis on limitations and failure modes: The paper does not explicitly discuss cases where CREAT3R might underperform, such as scenes with repetitive textures, highly reflective surfaces, or extreme lighting variations. An analysis of such failure cases would provide valuable insights into the robustness and generalizability of the approach, as well as guidance for future improvements or hybrid strategies. | $\triangleright$ As suggested, a comprehensive Limitations and Discussion section (**Section 5**) has been added. We specifically discuss outward-facing scenarios, where candidate views share minimal overlap with the estimated geometry. We note that this vulnerability is inherent to the active selection paradigm, affecting all baselines similarly. To address this, we implement a regularization strategy that temporarily masks such candidates from the pool until the confidence field expands sufficiently to establish geometric connectivity. |
> | **Weakness 2.** Insufficient empirical evaluation of efficiency claims: Although the method claims improved computational efficiency and reduced optimization time, the experimental section lacks a detailed quantitative analysis of runtime, memory consumption, or scalability compared to existing methods. Reporting metrics such as per-iteration runtime, GPU memory usage, and end-to-end reconstruction time would substantiate the paper’s efficiency claims and better demonstrate its practical advantages for real-world deployment. | $\triangleright$ Due to space constraints, we have included a detailed per-iteration runtime comparison in the appendix. As summarized in Table 5, running on a single V100 GPU, Creat3r takes an average of 10 seconds per iteration, significantly faster than the baselines, which require 24–40 seconds. |
>
> #### Part 2: Questions
> | Question/Comment | Author Response |
> | :--- | :--- |
> | **Question 1.** Pose acquisition without SfM: The paper emphasizes that CREAT3R avoids Structure-from-Motion (SfM) to prevent information leakage from unseen views, yet it still requires known camera poses for all candidate images. Could the authors clarify how these poses are obtained in practice without relying on global SfM or similar reconstruction pipelines? Are the poses assumed to come from external sensors (e.g., IMU, SLAM, or GPS-based localization)? If so, how sensitive is CREAT3R to pose noise, and have the authors evaluated the robustness of the framework under pose perturbations or inaccuracies? | $\triangleright$ In practical active view selection scenarios (e.g., robotics), the "candidate poses" represent the reachable workspace or a pre-planned trajectory. The robot knows its coordinate system (via SLAM or odometry) and evaluates potential future poses. Creat3r selects the next-best view from this pre-defined pool. While the integration of noisy inertial data (IMU) for real-time localization is a critical engineering challenge, it falls outside the primary scope of this paper, which focuses on the selection strategy assuming a calibrated workspace. |
> | **Question 2.** Initialization strategy for known views: The paper mentions that the process begins with “a small set of known views,” but the method for selecting or initializing these views is not clearly described. Are the initial views chosen randomly, uniformly distributed in pose space, or based on heuristic criteria such as viewpoint diversity or coverage? Given that initialization can significantly influence subsequent selection quality, providing either an ablation or a justification for the initialization strategy would help clarify the reproducibility and stability of the approach. | $\triangleright$ The initialization strategy is detailed in Appendix A.1. To ensure fair comparison and reproducibility, we adhere to the standard practice established by the ReconFusion benchmark, using the specific set of three initial views defined in their protocol. This ensures that performance differences are due to the selection strategy rather than random variations in initialization. |

---

> ### Author Response · Authors · 2025-12-02
> **Responses to Reviewer KrRH**
>
> #### Part 1: Weaknesses
> | Question/Comment | Author Response |
> | :--- | :--- |
> | **Weakness 1.** The wording of the paper is imprecise. The method name ‘Creat3r’ is not aligned with its handling task. 3R models are for feed-forward 3D reconstructions, while this method is for active view selection. The definition of ‘confidence’ is also different from 3R models. | $\triangleright$ We respectfully clarify that the term "3R" (3D Reconstruction) is a general descriptor and is not strictly constrained to feed-forward methods in the literature. For instance, 3R-GS [1] proposes an optimization-based (not feed-forward) method for camera pose estimation. [1] Zhisheng Huang, Peng Wang, Jingdong Zhang, Yuan Liu, Xin Li, and Wenping Wang. 3R-GS: Best Practice in Optimizing Camera Poses Along With 3DGS. arXiv preprint arXiv:2504.04294 (2025). |
> | **Weakness 2.** The camera poses of candidate views are freely available information. This is because reconstructing the poses of the selected sparse view effectively reconstructs the poses of the candidate views simultaneously. To verify information leakage, the authors should only recover the poses from the selected sparse view and train 3DGS. | $\triangleright$ There appears to be a misunderstanding of the active vision setting. In this problem setup, the candidate camera poses are defined a priori (i.e., the search space of camera poses is known). Creat3r selects the next-best view based on the content of the currently known set and the geometry of the potential candidate poses. We do not "recover" candidate poses from the sparse view; instead, we evaluate which of the known available poses should be selected next in order to acquire the corresponding visual content from the selected viewing direction. |
> | **Weakness 3.** The NVS precision of comparative methods are questionable. The precision of original FisherRF before adapted to 3DGS should be reported. The precision of Lyu et al. is significantly lower than reported. | $\triangleright$ As noted in our paper (Lines 75 and 357), evaluation metrics reported in previous literature are fundamentally biased because those methods are initialized using a global SfM point cloud derived from all images (including the supposedly "unseen" views). Our reported results differ because we implement a rigorous, leakage-free protocol that relies solely on camera poses, excluding the image content prior to active view selection. The performance drop observed in the baselines (e.g., Lyu et al.) in our experiments directly reflects the removal of these illicit point-cloud priors derived from visual content. For FisherRF, we maintained consistency by adapting it to the same 3DGS framework used by Lyu et al. to ensure a unified comparison. |
>
> #### Part 2: Comments & Responses
> | Question/Comment | Author Response |
> | :--- | :--- |
> | **Comment 1.** Low Reconstruction Accuracy (PSNR < 20): | $\triangleright$ _Context of Sparse View Reconstruction:_ Our evaluation strictly follows the benchmark by Lyu et al., utilizing only 10 and 20 reconstruction views. This places our work squarely in the sparse view reconstruction domain, which inherently yields lower quality metrics than dense view scenarios. _Performance is Consistent with SOTA:_ In this challenging sparse setting, achieving PSNRs in the sub-20 range is common. Methods like FSGS[2] and CoR-GS[3], when trained with a similar sparse view count, report comparable PSNR scores. This validates that our performance is competitive and representative of the current state-of-the-art under these stringent constraints. [2] Zhu et al. "Fsgs: Real-time few-shot view synthesis using gaussian splatting." ECCV 2024 [3] Zhang et al. "Cor-gs: sparse-view 3d gaussian splatting via co-regularization." ECCV 2024 |
> | **Comment 2.** Active Learning Formulation (SfM vs. 3DGS Bottleneck): | $\triangleright$ Active Learning (AL) and Structure-from-Motion (SfM) are fundamentally incompatible due to their process requirements. AL relies on an iterative and dynamic acquisition loop where images are acquired one by one or in small batches. In contrast, robust SfM requires a large, static batch of images to ensure sufficient feature correspondences and stable geometric initialization, which directly conflicts with the AL paradigm. |

---

> ### Author Response · Authors · 2025-12-02
> **Problem Setting of Active 3D Reconstruction**
>
> In the active view selection setting, the camera poses for a candidate pool are defined *a priori*, while the corresponding image content remains inaccessible until a specific camera pose is selected. Once a pose is chosen, the "*view*" (image) can then be "*captured*" and thus *exposed* to the view-selection algorithm. The process involves iteratively selecting the next viewing direction (camera pose) for acquiring the best view (image) from the candidate viewing pool. The selected view can then be added to the "*seen*" set to optimize the 3DGS model (baseline methods).
>
> ---
>
> &ensp;&ensp; **Initial State**:  Candidate Pose Pool { (`Pose Known`, `Image Content Unknown`)$_n$ | $n = 1,...,N$ } \
> &ensp;&ensp; **Iteration**: \
> &ensp; &ensp; &ensp; &ensp; *i*. Algorithm Predicts Best Next Pose $k$ from Candidate Pool \
> &ensp; &ensp; &ensp; &ensp; &ensp;        (Selection Made)                          \
> &ensp; &ensp; &ensp; &ensp; *ii*. "*Capture/Expose*" Image of View $k$ (Content Revealed from Pool) \
> &ensp; &ensp; &ensp; &ensp; &ensp;        (New Information Gained)                  \
> &ensp; &ensp; &ensp; &ensp; *iii*. Add View $k$ to "*Seen*" Set & Optimize 3DGS Model
>
> ---
>
> Therefore, given a candidate pool of camera poses (where the algorithm can acquire the corresponding images), the algorithm needs to predict which camera pose/viewing direction is more likely to obtain an informative image to improve 3DGS reconstruction.
>
> In such a setting, the benchmark established by Lyu et al. attempts to simulate this scenario using the Mip-NeRF 360 dataset, revealing pixel data only after selection to mimic an autonomous agent. However, their methodology introduces a critical flaw: it utilizes a sparse **point cloud** initialized via a global Structure-from-Motion (SfM) pre-processing step that incorporates **images** from all candidate poses. This practice constitutes significant information leakage, as it implicitly provides the algorithm with geometric priors derived from "**supposedly unseen**" **image data**, thereby contradicting the fundamental constraints of active vision.
>
> To address this, we refine the benchmark by replacing the global SfM point-cloud prior with rigorous, leakage-free initialization protocols. In the iterative process, we also decouple 3DGS optimization from selection by introducing robust point estimation and confidence reaggregation.

---

### Note · Authors · 2026-01-30

I have read and agree with the venue's withdrawal policy on behalf of myself and my co-authors.

---

### Meta-Review · Area_Chair_KpKJ · 2026-01-07

**Summary:**

There are several major concerns raised by the reviewers:
- Sparse View Setting: only 10 to 20 view reconstruction quality is evaluated, the asymptotic setting (dense view high quality reconstruction) is not considered.
- Information Leakage: Regarding how the camera poses should be revealed to the active reconstruction algorithm to guarantees fairness and connection to real-world settings.
- Experimental Results: Reviewers suggest that some numbers are not inconsistent with the ones reported in the papers.

**Reviewer Concerns:**

- The sparse view setting issue is explained by not fully addressed. Evaluating the geometric and visual quality of the 3D reconstruction only based on 10 to 20 views limits the application of the proposed method from high quality reconstruction tasks.
- The information leakage issue is addressed by the author via problem setting clarification.
- The inconsistent experimental numbers is not fully addressed. The experiments are conducted on a slightly different experimental setting.

**Reviewer Scores:**

I expect reviewer KrRH to raise the score to 4 and no other change of score.

---

### Decision · Program_Chairs · 2026-01-26

Reject